# STYLE SPECTROSCOPE: IMPROVE INTERPRETABILITY AND CONTROLLABILITY THROUGH FOURIER ANALYSIS

## ABSTRACT

Universal style transfer (UST) infuses styles from arbitrary reference images into content images. Existing methods, while enjoying many practical successes, are unable of explaining experimental observations, including different performances of UST algorithms in preserving the spatial structure of content images. In addition, methods are limited to cumbersome global controls on stylization, so that they require additional spatial masks for desired stylization. In this work, we provide a systematic Fourier analysis on a general framework for UST. We present an equivalent form of the framework in the frequency domain. The form implies that existing algorithms treat all frequency components and pixels of feature maps equally, except for the zero-frequency component. We connect Fourier amplitude and phase with Gram matrices and a content reconstruction loss in style transfer, respectively. Based on such equivalence and connections, we can thus interpret different structure preservation behaviors between algorithms with Fourier phase. Given the interpretations we have, we propose two manipulations in practice for structure preservation and desired stylization. Both qualitative and quantitative experiments demonstrate the competitive performance of our method against the state-of-the-art methods. We also conduct experiments to demonstrate (1) the abovementioned equivalence, (2) the interpretability based on Fourier amplitude and phase and (3) the controllability associated with frequency components.

## 1 INTRODUCTION

Style transfer deals with the problem of synthesizing an image which has the style characteristics from a style image and the content representation from a content image. The seminal work (Gatys et al., 2016) uses Gram matrices of feature maps to model style characteristics and optimizes reconstruction losses between the reference images and stylized images iteratively. For the purpose of gaining vivid visual styles and less computation cost, more trained feed-forward networks are proposed (Wang et al., 2020; Li & Wand, 2016; Johnson et al., 2016; Sheng et al., 2018; Li et al., 2017b; Sty; Dumoulin et al., 2017). Recent works focus on arbitrary style transfer (Park & Lee, 2019; Chen et al., 2021a;b; Chandran et al., 2021), or artistic style (Chen et al., 2021b; Liu et al., 2021; Chen et al., 2021c). These works capture limited types of style and cannot well generalize to unseen style images (Hong et al., 2021).

To obtain the generalization ability for arbitrary style images, many methods are proposed for the task of universal style transfer (UST). Essentially, the main challenge of UST is to properly extract the style characteristics from style images and transfer them onto content images without any prior knowledge of target style. The representative methods of UST consider various notions of style characteristics. For example, AdaIN (Huang & Belongie, 2017) aligns the channel-wise means and variances of feature maps between content images and style images, and WCT (Li et al., 2017a) further matches up the covariance matrices of feature maps by means of whitening and coloring processes, leading to more expressive colors and intensive stylization.

While these two approaches and their derivative works show impressive performances on stylization, they behave differently in preserving the structure of content images. It is observed that the operations performed by AdaIN can do better in structure preservation of content images while those

of WCT might introduce structural artifacts and distortions. Many follow-up works focus on alle-viating the problem of WCT (Li et al., 2018; Chiu & Gurari, 2022; Yoo et al., 2019), but seldom can analytically and systematically explain what makes the difference. In the field of UST, we need an analytical theory to bridge algorithms with experimental phenomena for better interpretability, potentially leading to better stylization controls. To this end, we resort to apply Fourier transform for deep analysis, aiming to find new equivalence in frequency domain and bring new interpretations and practical manipulations to existing style transfer methods.

In this work, we first revisit a framework by (Li et al., 2017a) which unifies several well-known UST methods. Based on the framework, we derive an equivalent form for it in the frequency domain, which has the same simplicity with its original form in the spatial domain. Accordingly, the derived result demonstrates that these UST methods perform a uniform transformation in the frequency domain except for the origin. Furthermore, these UST methods transform frequency components (excluding the zero-frequency component) and spatial pixels of feature maps in an identical manner. Thus, these UST methods perform manipulations on the whole frequency domain instead of specific subsets of frequencies (either high frequencies or low frequencies).

Secondly, through the lens of the Fourier transform, we further explore the relation of Fourier phase and amplitude with key notions in style transfer, and then we present new interpretations based on the equivalence we have. On one hand, we prove that a content reconstruction loss between two feature maps reaches a local minimum when they have identical Fourier phase, which implies that Fourier phase of feature maps contributes to the structure of stylized results. On the other hand, we prove that the Fourier amplitude of feature maps determines the diagonals of their Gram matrices, which implies that Fourier amplitude contributes to the intensity information of stylized images. Next, We demonstrate that AdaIN does preserve the Fourier phase of feature maps while WCT does not, and we interpret the different behaviors between the UST methods in structure preservation as a consequence of their different treatment with the Fourier phase of feature maps.

Thirdly, based on the connection we establish between style transfer and Fourier transfer, we propose two manipulations on the frequency components of feature maps: 1) a phase replacement operation to keep phase of feature maps unchanged during stylization for better structure preservation. 2) a feature combination operation to assign different weights to different frequency components of fea-ture maps for desired stylization. We then conduct extensive experiments to validate their efficacy.

The contributions of this paper are summarized as follows:

- **Equivalence** We present a theoretically equivalent form for several state-of-the-art UST methods in the frequency domain and reveal their effects on frequencies. We conduct corresponding experiments to validate the equivalence.

- **Interpretability** We connect Fourier amplitude and phase with key notions in style transfer and present new interpretations on different behaviors of UST methods. The interpretations are validated by experiments.

- **Controllability** We propose two manipulations for structure preservation and desired styl-ization. We have experimental validation for their efficacy and controllability.

## 2 PRELIMINARIES

### 2.1 FOURIER TRANSFORM

The Fourier transform has been widely used for the analysis of the frequency components in signals, including images and feature maps in the shallow layers of neural networks. Given an image $F \in \mathbb{R}^{C \times H \times W}$, the discrete Fourier transform (DFT) (Jenkins & Desai, 1986) decomposes it into a unique representation $\mathcal{F} \in \mathbb{C}^{C \times H \times W}$ in the frequency domain as follows:

$$\mathcal{F}_{u,v} = \sum_{h=0}^{H-1} \sum_{w=0}^{W-1} F_{h,w} e^{-j2\pi(u\frac{h}{H} + v\frac{w}{W})}, \quad j^2 = -1, \tag{1}$$

where $(h, w)$ and $(u, v)$ are the indices on the spatial dimensions and the frequency dimensions, respectively. Since images and feature maps consist of multiple channels, we here apply the Fourier

transform upon each channel separately and omit the explicit notation of channels. Each frequency component $\mathcal{F}_{u,v}$ can be decomposed into amplitude $|\mathcal{F}_{u,v}|$ and phase $\angle\mathcal{F}_{u,v}$:

$$|\mathcal{F}_{u,v}| = \sqrt{(\mathcal{R}_{u,v})^2 + (\mathcal{I}_{u,v})^2}, \quad \angle\mathcal{F}_{u,v} = \mathtt{atan2}(\mathcal{I}_{u,v}, \mathcal{R}_{u,v}), \tag{2}$$

where $\mathcal{R}_{u,v}$ and $\mathcal{I}_{u,v}$ are the real part and the imaginary part of the complex frequency component $\mathcal{F}_{u,v}$, respectively. Intuitively, as for images, amplitude carries much of intensity information, including the contrast or the difference between the brightest and darkest peaks of images, and phase crucially determines the spatial content of images (Gonzalez & Woods, 2008).

## 2.2 A UNIFIED FRAMEWORK FOR UNIVERSAL STYLE TRANSFER

To better demonstrate the connection between style transfer and the Fourier transform, a unified framework of different style transfer methods is preferred to serve as a bridge. Given a content image $I^c$ and a style image $I^s$, we denote the feature maps of $I^c$ and $I^s$ as $F^c \in \mathbb{R}^{C \times H^c \times W^c}$ and $F^s \in \mathbb{R}^{C \times H^s \times W^s}$ respectively, where $C$ denotes the number of channels, $H^c$ ($H^s$) the height and $W^c$ ($W^s$) the width. For a majority of UST methods, their goal is to transform the content image feature maps $F^c$ into stylized feature maps $F^{cs}$, whose first-order and second-order statistics are aligned with those of the style image feature maps $F^s$. Accordingly, their methods mainly depend on the corresponding channel-wise mean vectors $\mu^c, \mu^s \in \mathbb{R}^C$ and the covariance matrices $\Sigma^c, \Sigma^s \in \mathbb{R}^{C \times C}$ of $F^c$ and $F^s$, respectively.

A framework is proposed in (Lu et al., 2019) for unifying several well-known methods under the same umbrella. Specifically, each pixel $F_{h,w}^c$ of $F^c$ is first centralized by subtracting the mean vector $\mu^c$, where $h$ and $w$ are indices on spatial dimensions. Then the framework linearly transforms $F_{h,w}^c$ with the transformation matrix $T \in \mathbb{R}^{C \times C}$ and re-centers $F_{h,w}^c$ by adding the mean vector $\mu^s$ of the style. Each pixel $F_{h,w}^{cs} \in \mathbb{R}^C$ of stylized feature maps can be represented as follows:

$$F_{h,w}^{cs} = T\left(F_{h,w}^c - \mu^c\right) + \mu^s, \tag{3}$$

where the transformation matrix $T$ has multiple forms based on a variety of configurations of different methods. We here demonstrate the relation between the unified framework and several methods in details.

1. **AdaIN** In *Adaptive Instance Normalization* (AdaIN) (Huang & Belongie, 2017), the transformation matrix $T = \mathtt{diag}\left(\Sigma^s\right)/\mathtt{diag}\left(\Sigma^c\right)$, where $\mathtt{diag}\left(\Sigma\right)$ denotes the diagonal matrix of a given matrix $\Sigma$ and $/$ denotes the element-wise division. Because of the characteristics of diagonal matrices, only the means and variances within each single feature channel of $F^{cs}$ are matched up to those of $F^s$, ignoring the correlation between channels.

2. **WCT** Instead of shifting a single set of intra-channel statistics, (Li et al., 2017a) proposes a *Whitening and Coloring Transform* (WCT) that focuses further on the alignment of covariance matrices. Similar with AdaIN, the transformation matrix for WCT is $T = (\Sigma^s)^{\frac{1}{2}}(\Sigma^c)^{-\frac{1}{2}}$, leading to well-aligned second-order statistics.

3. **LinearWCT** While WCT generates stylized images more expressively, it is still computationally expensive because of the high dimensions of feature maps in neural networks. (Li et al., 2019) proposes LinearWCT to use light-weighted neural networks to model the linear transformation $T$ by optimizing the Gram loss, known as a widely-used style reconstruction objective function:

$$T = \underset{T}{\arg\min}\|F^{cs}F^{cs\top} - F^sF^{s\top}\|_F^2, \tag{4}$$

where $F^{cs}F^{cs\top}$ is the Gram matrix for $F^{cs}$ and $\|\cdot\|_F^2$ denotes the squared Frobenius norm of the differences between given matrices.

4. **OptimalWCT** Similarly, (Lu et al., 2019) proposes OptimalWCT to derive a closed-form solution for $T$ without the help of optimization process:

$$T = (\Sigma^c)^{-\frac{1}{2}}\left((\Sigma^c)^{\frac{1}{2}}\Sigma^s(\Sigma^c)^{\frac{1}{2}}\right)^{\frac{1}{2}}(\Sigma^c)^{-\frac{1}{2}}. \tag{5}$$

Their method reaches the theoretical local minimum for the content loss $\mathcal{L}_c = \|F^c - F^{cs}\|_F^2$, which is widely-used in style transfer (Huang & Belongie, 2017; Gatys et al., 2016; Lu et al., 2019) for structure preservation of content images.

## 3 METHOD

In this section, we first show an equivalent form of the framework in the frequency domain. In this way, all the methods based on the framework in Section 2.2 can be interpreted as effecting on the frequency domain. We further connect amplitude and phase with existing concepts in the context of style transfer, and explain why WCT might not preserve the structure of content images. Finally, we propose two operations for better structure preservation and desired stylization.

### 3.1 THE EQUIVALENT FORM OF THE FRAMEWORK IN THE FREQUENCY DOMAIN

We theoretically analyze the unified framework from the angle of 2-D DFT. We denote the DFT of $F^{cs}$ as $\mathcal{F}^{cs} \in \mathbb{C}^{C \times H^c \times W^c}$, where $\mathbb{C}$ is the set of complex numbers. According to the unified framework in Eq. (13), we can derive each complex frequency component $\mathcal{F}^{cs}_{u,v}$ as:

$$
\begin{aligned}
\mathcal{F}^{cs}_{u,v} &= \sum_{h=0}^{H^c-1} \sum_{w=0}^{W^c-1} F^{cs}_{h,w} e^{-j2\pi(u\frac{h}{H^c}+v\frac{w}{W^c})} = \sum_{h=0}^{H^c-1} \sum_{w=0}^{W^c-1} [T(F^c_{h,w} - \mu^c) + \mu^s] e^{-j2\pi(u\frac{h}{H^c}+v\frac{w}{W^c})} \\
&= T \underbrace{\sum_{h=0}^{H^c-1} \sum_{w=0}^{W^c-1} F^c_{h,w} e^{-j2\pi(u\frac{h}{H^c}+v\frac{w}{W^c})}}_{\begin{cases} H^cW^c\mu^c, & \text{if } u=v=0; \\ \mathcal{F}^c_{u,v}, & \text{else} \end{cases}} + [\mu^s - T\mu^c] \underbrace{\sum_{h=0}^{H^c-1} \sum_{w=0}^{W^c-1} e^{-j2\pi(u\frac{h}{H^c}+v\frac{w}{W^c})}}_{\begin{cases} H^cW^c, & \text{if } u=v=0; \\ 0, & \text{else} \end{cases}} \\
&= \begin{cases} H^cW^c\mu^s, & \text{if } u=v=0; \\ T\mathcal{F}^c_{u,v}, & \text{else} \end{cases} = \begin{cases} \left(\frac{H^cW^c}{H^sW^s}\right)\mathcal{F}^s_{0,0}, & \text{if } u=v=0; \\ T\mathcal{F}^c_{u,v}, & \text{else} \end{cases},
\end{aligned}
$$
(6)

where $u$ and $v$ are indices upon the frequency dimensions, and $\mathcal{F}^c$ and $\mathcal{F}^s$ are the DFTs of $F^c$ and $F^s$, respectively. According to the Fourier transform in Eq. (1), $\mathcal{F}_{0,0} = \sum_{h=0}^{H-1} \sum_{w=0}^{W-1} F_{h,w} = HW\mu$. Thus, we have $\mathcal{F}^{cs}_{u,v} = H^cW^c\mu^s = \left(\frac{H^cW^c}{H^sW^s}\right)\mathcal{F}^s_{0,0}$ when $u = v = 0$. Therefore, in the frequency domain, style transfer methods based on the unified framework are simple linear transformations on $\mathcal{F}^c$ except for the zero-frequency component $\mathcal{F}^c_{0,0}$, which is replaced with the re-scaled zero-frequency component of $\mathcal{F}^s$.

From Eq.(6), we find that each individual frequency component (excluding the zero-frequency component) has an identical linear transformation with pixels on the feature maps. In this way, there is no entanglement between different frequencies in the process of style transfer. Thus, it is feasible to treat and manipulate each frequency component of $\mathcal{F}^{cs}$ as an individual for practical usage. Therefore, we justify the claim that mainstream methods in Section 2.2 for UST are not sole transfer on specific subsets of frequencies (either high frequencies or low frequencies), but essentially on the whole frequency domain.

### 3.2 CONNECTIONS AND INTERPRETATIONS: AMPLITUDE AND PHASE

To better bridge style transfer with the Fourier transform, we connect phase and amplitude with a reconstruction loss and the Gram matrix in style transfer, respectively.

**Phase and the content loss** We here demonstrate the relation between phase and the content loss, which widely serves as a construction loss for optimizing the differences of spatial arrangement between stylized images $I^{cs}$ and content images $I^c$. Given their feature maps $F^{cs}, F^c \in \mathbb{R}^{C \times H \times W}$, corresponding DFTs $\mathcal{F}^{cs}, \mathcal{F}^c$, Fourier amplitude $|\mathcal{F}^{cs}|, |\mathcal{F}^c|$ and Fourier phase $\angle\mathcal{F}^{cs}, \angle\mathcal{F}^c$, the content loss between $F^{cs}$ and $F^c$ can be derived as:

$$
\begin{aligned}
\mathcal{L}_c &= \|F^{cs} - F^c\|_F^2 = \sum_{k=0}^{C} \sum_{h=0}^{H} \sum_{w=0}^{W} \left(F^{cs}_{k,h,w} - F^c_{k,h,w}\right)^2 = \frac{1}{HW} \sum_{k=0}^{C} \sum_{u=0}^{H} \sum_{v=0}^{W} |\mathcal{F}^{cs}_{k,u,v} - \mathcal{F}^c_{k,u,v}|^2 \\
&= \frac{1}{HW} \sum_{k=0}^{C} \sum_{u=0}^{H} \sum_{v=0}^{W} \left[ |\mathcal{F}^{cs}_{k,u,v}|^2 + |\mathcal{F}^c_{k,u,v}|^2 - 2|\mathcal{F}^{cs}_{k,u,v}||\mathcal{F}^c_{k,u,v}| \cos\left(\angle\mathcal{F}^{cs}_{k,u,v} - \angle\mathcal{F}^c_{k,u,v}\right) \right],
\end{aligned}
$$
(7)

where the second equality is held by the Parseval's theorem and $k$, $(h, w)$ and $(u, v)$ are indices on channels, spatial dimensions and frequency dimensions, respectively. When $F^{cs}$ is optimized for the content loss, since $|\mathcal{F}^{cs}_{k,u,v}|$ and $|\mathcal{F}^c_{k,u,v}|$ are non-negative numbers, the content loss $\mathcal{L}_c$ reaches a local minimum when $\angle\mathcal{F}^{cs}_{k,u,v} = \angle\mathcal{F}^c_{k,u,v}$ for all $(k, u, v)$. Furthermore, whenever $\angle\mathcal{F}^{cs}_{k,u,v}$ gets closer to $\angle\mathcal{F}^c_{k,u,v}$, the content loss decreases, demonstrating the crucial role of phase of feature maps in determining the spatial information of corresponding decoded images. Therefore, we can interpret the structure preservation abilities of methods from the perspective of Fourier phase. Furthermore, we can manipulate Fourier phase for better performances in structure preservation.

**Interpretations on structure preservation**    Based on the equivalent form in Eq. (6) and the relation between Fourier phase and the content loss, we can give interpretations to different behaviors of methods in structure preservation. Concerning AdaIN and WCT as instances of the equivalent framework in the frequency domain, we have $\mathcal{F}^{cs}_{u,v} = T\mathcal{F}^c_{u,v}$ when $(u, v) \neq (0, 0)$. Note that for AdaIN, the transformation matrix $T = \texttt{diag}\,(\Sigma^s)\,/\texttt{diag}\,(\Sigma^c) \in \mathbb{R}^{C \times C}$ is a real diagonal matrix, which has the same scaling upon the real part and the imaginary part of $\mathcal{F}^c$. As a result, AdaIN can preserve the phase in each feature channel and keep the content loss of feature maps in a local minimum. While WCT provides a non-diagonal matrix $T = (\Sigma^s)^{\frac{1}{2}}(\Sigma^c)^{-\frac{1}{2}}$ for transformation, the information between different channels is consequently entangled, the phase of each channel is disturbed and the content loss after the process of WCT is likely to increase much more than the one after the process of AdaIN. Therefore, WCT needs more efforts to preserve the spatial information of content images, resulting in its less appealing performances in structure preservation.

**Amplitude and Gram matrices**    We theoretically demonstrate the connection between the Fourier amplitude of feature maps and their Gram matrices. Given feature maps $F \in \mathbb{R}^{C \times H \times W}$, corresponding Fourier amplitude $|\mathcal{F}|$ and Fourier phase $\angle\mathcal{F}$ of their DFT $\mathcal{F}$, the pixels of the Gram matrix $FF^\top$ can be derived as:

$$
\begin{aligned}
\left(FF^\top\right)_{c_1,c_2} &= \sum_{h=0}^{H}\sum_{w=0}^{W} F_{c_1,h,w}F_{c_2,h,w} = \frac{1}{HW}\sum_{u=0}^{H}\sum_{v=0}^{W}\mathcal{F}_{c_1,u,v}\mathcal{F}^*_{c_2,u,v} \\
&= \frac{1}{HW}\sum_{u=0}^{H}\sum_{v=0}^{W}|\mathcal{F}_{c_1,u,v}||\mathcal{F}_{c_2,u,v}|\cos\left(\angle\mathcal{F}_{c_1,u,v} - \angle\mathcal{F}_{c_2,u,v}\right),
\end{aligned}
\tag{8}
$$

where $c_1, c_2$ are indices on the channels, $(*)$ represents complex conjugate and the second equality is held by the Parseval's theorem. Since $\left(FF^\top\right)_{c_1,c_2}$ is a real number, we omit the imaginary part in the final step. In a special case where $c_1$ equals $c_2$, $\left(FF^\top\right)_{c_1,c_2}$ equals $\frac{1}{HW}\sum_{u=0}^{H}\sum_{v=0}^{W}|\mathcal{F}_{c_1,u,v}|^2$. This indicates that the sum of the square of amplitude components directly determines the diagonals of Gram matrices. Each elements on the diagonal of Gram matrices represents infra-channel second-order statistics of feature maps, measuring the intensity of information in each channel. Therefore, if we only manipulate the Fourier phase of the DFTs of feature maps and keep the Fourier amplitude unchanged, it can be expected that the intensity presentations of the corresponding decoded images are roughly the same. For detailed proof, see the Supplementary Materials.

## 3.3   MANIPULATIONS ON STYLIZED FEATURE MAPS IN THE FREQUENCY DOMAIN

The equivalent form in Eq. (6) and abovementioned connections enable further manipulations for better structure preservation or desired stylization. We propose two simple operations upon frequency components of feature maps, which we call phase replacement and frequency combination.

**Phase replacement**    Given the DFT of the content feature maps $\mathcal{F}^c$ and the DFT of the stylized feature maps $\mathcal{F}^{cs}$, we calculate the phase of $\mathcal{F}^c$, denoted as $\angle\mathcal{F}^c \in [0, 2\pi)^{C \times H^c \times W^c}$ and the amplitude of $\mathcal{F}^{cs}$, denoted as $|\mathcal{F}|^{cs} \in \mathbb{R}^{C \times H^c \times W^c}_+$, where $\mathbb{R}_+$ denotes the set of non-negative real numbers. We then reconstruct $\mathcal{F}^{cs}$ as:

$$
\mathcal{F}^{cs}_{u,v} = |\mathcal{F}|^{cs}_{u,v} \odot \cos\angle\mathcal{F}^c_{u,v} + j|\mathcal{F}|^{cs}_{u,v} \odot \sin\angle\mathcal{F}^c_{u,v},
\tag{9}
$$

where $\cos$ and $\sin$ are element-wise operators on vectors (*e.g.*, $\cos\phi = [\cos\phi^1, ..., \cos\phi^C]$), $\odot$ is the element-wise multiplication and $j$ is the imaginary unit. Based on the connections established in Section 3.2, when we replace the phase of $\mathcal{F}^{cs}$ with $\angle\mathcal{F}^c$, the content loss between $F^{cs}$ and $F^c$

is reduced and in this way, the structure of content images is more preserved in $F^{cs}$. In addition, since the amplitude of $\mathcal{F}^{cs}$ is not changed, the diagonal of $F^{cs}F^{cs\top}$ stays unchanged and so does the basic intensity information of the stylized results. A similar amplitude transferring method is proposed in (Yang & Soatto, 2020) for semantic segmentation, which shares resembling views on phase and amplitude with ours.

**Frequency combination**    To accommodate different requirements from users, appropriate control on the stylization is needed for practical usage. Plenty of works for style transfer use linear combination of content feature maps $F^c$ and stylized feature maps $F^{cs}$ as shown in Eq. (10):

$$F^{cs} = \alpha F^{cs} + (1 - \alpha)F^c, \tag{10}$$

where $\alpha$ is the weight for controlling on the stylization. In this way, all the global characteristics of images (*e.g.*, the sharp edges of trees and the smooth background of sky) are combined uniformly. While in most cases, users are expecting for customized global changes on images (*e.g.*, having the details of trees less stylized but keeping the sky moderately stylized). Since high frequencies determine the details and low frequencies determine the overview of images, we can accommodate the customized needs of users with combination of frequencies in different proportions.

Given the DFT of content feature maps $\mathcal{F}^c$ and the DFT of stylized feature maps $\mathcal{F}^{cs}$, we first rearranges their frequency components with the zero-frequency components in the center point $(u_0, v_0)$, following a common technique in digital image processing. In this way, the frequency components close to $(u_0, v_0)$ are low-frequency components whereas the rest of components represent high frequencies. Next, we combine $\mathcal{F}^{cs}$ and $\mathcal{F}^c$ using a weighting function $\alpha : \mathbb{R}^2 \to [0, 1]$:

$$\mathcal{F}^{cs}_{u,v} = \alpha(u,v)\mathcal{F}^{cs}_{u,v} + [1 - \alpha(u,v)]\mathcal{F}^c_{u,v}, \tag{11}$$

where $\alpha$ serves as the stylization weighting function dependent on the indices $(u, v)$. For example, if users want to have the details less stylized, higher frequencies of $\mathcal{F}^{cs}$ need to be less weighted, and accordingly a lower value of $\alpha$ can be set for $(u, v)$ indexing higher frequencies. In practice, the function $\alpha$ is set to be controlled by a hyper-parameter $\sigma$:

$$\alpha(u,v) = \exp\left[-\frac{(u - u_0)^2 + (v - v_0)^2}{\sigma}\right], \tag{12}$$

where $\sigma$ represents the degree for combining the low frequencies of $\mathcal{F}^{cs}$. When $\sigma$ gets larger, the value of $\alpha(u,v)$ increases for every $(u, v)$. In this way, more low frequencies of $\mathcal{F}^{cs}$ (indexed by $(u, v)$ close to $(u_0, v_0)$ ) are gradually kept.

## 4    EXPERIMENTS

In this section, we first introduce our method specification and implementation details. Then we compare our method with the state-of-the-art style transfer methods in terms of visual effect, structure preservation and computing time. Moreover, we conduct experiments to validate the equivalence presented in Eq. (6), the interpretations on Fourier amplitude and phase introduced in Section 3.2, and the efficacy of manipulations proposed in Section 3.3. More qualitative results and implementation details are available in the Supplementary Materials.

**Method specification.**    Based on the equivalence and connections mentioned above, the proposed method first performs a selected UST algorithm in the frequency domain. In practice, we choose to implement our method in conjunction with WCT because WCT produces expressive stylization in spite of introduced distortions. To deal with the distortions, we adopt phase replacement (PR) to substitute the Fourier phase of stylized feature maps with that of content feature maps. Since PR can optimize the content loss to a local minimum according to Eq. (7), the structure of content images is preserved. Finally, the proposed method uses inverse discrete Fourier transform to reverse the frequency components back to the spatial domain.

**Implementation details.**    We adopt a part of the VGG-19 network (Simonyan & Zisserman, 2015) (from the layer *conv*1_1 to the layer *conv*4_1) as our encoder. The weights of our encoder are borrowed from ImageNet-pretrained weights, following existing style transfer methods. We train our decoder for image reconstruction by minimizing the $L_2$ reconstruction loss. During the

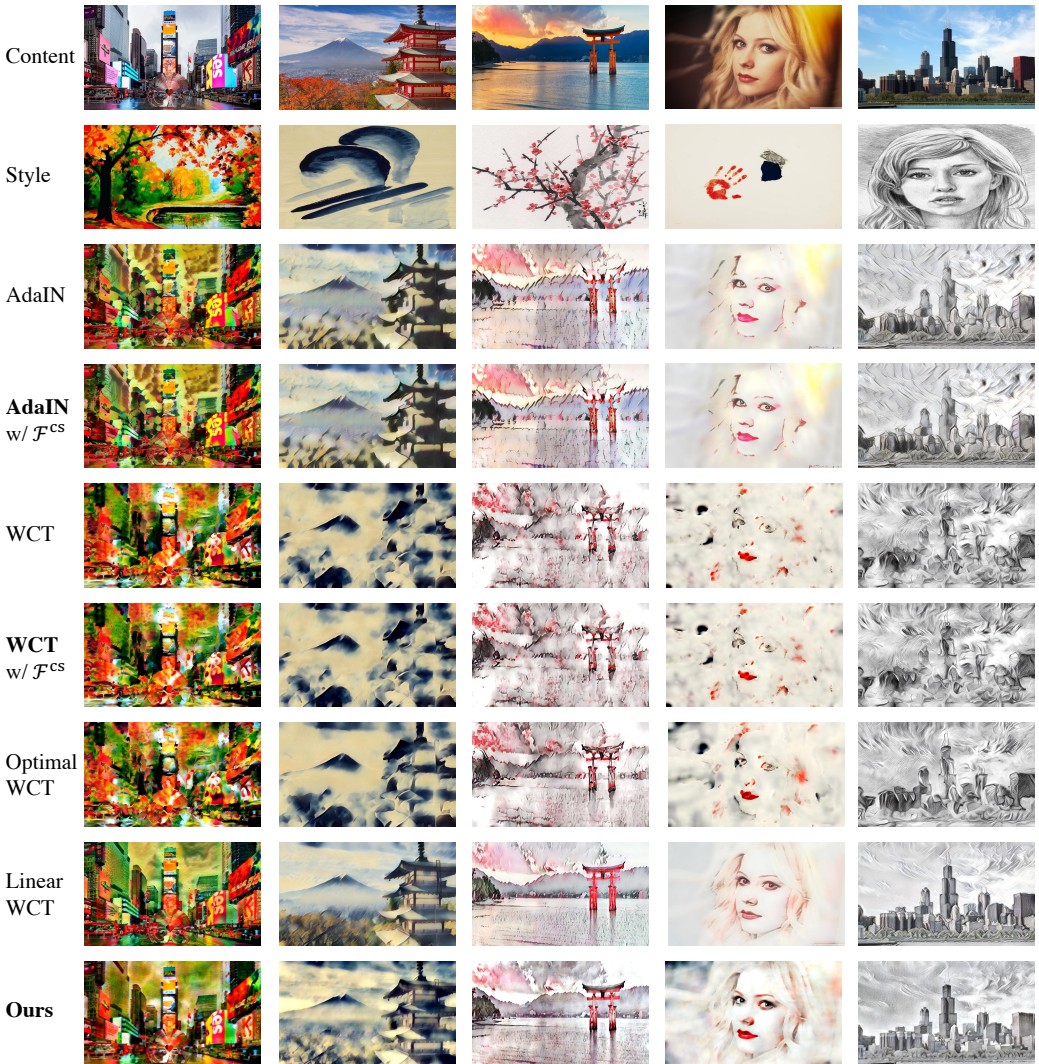

Figure 1: Qualitative comparison on the state-of-the-art UST algorithms. AdaIN w/ $\mathcal{F}^{cs}$ and WCT w/ $\mathcal{F}^{cs}$ are implemented following Eq. (6) in the frequency domain, which perform equivalent stylization with the original methods.

Table 1: The SSIM scores, the Gram loss and the time cost (seconds) for different UST methods.

| Method | AdaIN | WCT | LinearWCT | OptimalWCT | AvatarNet | SANet | Self-Contained | Ours |
|---|---|---|---|---|---|---|---|---|
| SSIM | 0.307 | 0.234 | 0.378 | 0.250 | 0.329 | 0.310 | 0.276 | **0.403** |
| SSIM (w/ PR) | 0.307 | 0.251 | 0.427 | 0.263 | - | - | - | **0.438** |
| Time | **0.0218** | 0.3167 | 0.0038 | 0.6247 | 3.027 | 0.0052 | 0.0513 | 0.098 |
| Time (w/ PR) | **0.0226** | 0.3293 | 0.0045 | 0.6321 | - | - | - | 0.112 |

inference stage, we apply our method to feature maps in each layer of the decoder. We choose MS-COCO dataset (Lin et al., 2014) and WikiArt dataset (Nichol, 2016) as our content dataset and style dataset, respectively. Our decoder is also trained on the content dataset, whose training images are first resized into $512{\times}512$ and randomly cropped into a size of $256{\times}256$. We run all the experiments on a single NVIDIA Tesla V100.

## 4.1 PERFORMANCE COMPARISON

**Qualitative comparison** In Figure 1, we show some visualization results of the qualitative comparison between the proposed methods and the state-of-the-art UST methods (*i.e.*, AdaIN (Huang

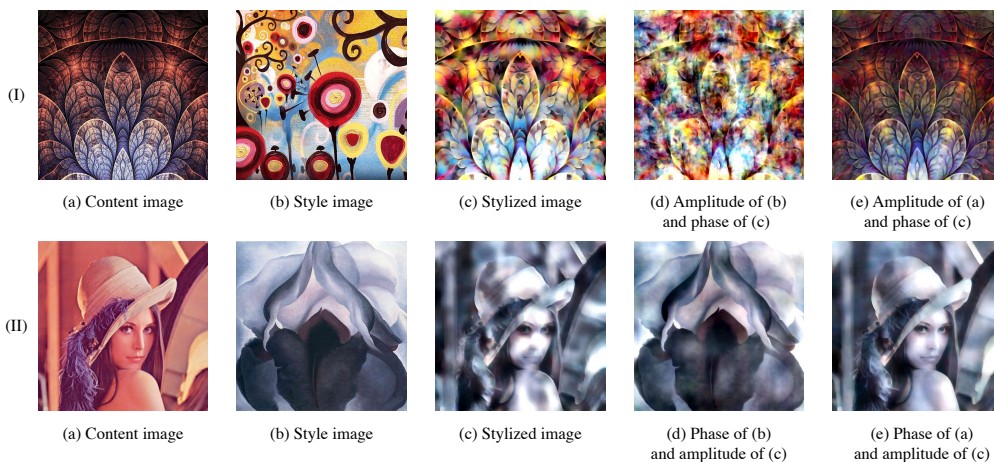

Figure 2: Synthesized results with replaced Fourier amplitude or phase.

& Belongie, 2017), WCT (Li et al., 2017a), LinearWCT (Li et al., 2019), OptimalWCT (Lu et al., 2019), SANet(Park & Lee, 2019), AvatarNet(Sheng et al., 2018) and Self-Contained (Chen et al., 2020)). We observe that AdaIN roughly preserves the structure of images, but often produces unappealing patterns on the edges (*e.g.*, $2^{nd}$, $3^{rd}$, and $5^{th}$ columns). WCT and OptimalWCT can produce intensive but distorted artistic style and yield images less similar with content images in structure (*e.g.*, $2^{nd}$, $4^{th}$, and $5^{th}$ columns). LinearWCT roughly preserves the spatial structure of content images, but the stylization is less intensive (*e.g.*, $1^{st}$, $2^{nd}$, and $4^{th}$ columns). Comparatively, the proposed method performs well in the contrast and intensity of stylized results (*e.g.*, the misty sky and the red lips). The proposed method also well preserves the spatial structure of content images, including the details (*e.g.* the contours of architectures and hair) and the overview (*e.g.* the light and shadow of cloudy sky and the human face) of images.

**Quantitative comparison**     In addition to the comparison on visual effect, we conduct a quantitative comparison. Moreover, we implement PR on each method and present corresponding quantitative scores. The Structural Similarity Index (SSIM) between the stylized images and corresponding content images is adopted as the metric for evaluating structure preservation. In Table 1, our method achieves the highest SSIM score and PR improves SSIM scores of all methods except for AdaIN, since AdaIN does not change the Fourier phase of feature maps during stylization. The improved SSIM validates our interpretations on structure preservation behaviors between alogrithms. Regarding the computing time, while our method needs to utilize Fourier transform, it still has a competitive time cost compared with other methods. It is noteworthy that our method has less additional computational cost if PR is adopted, since it has already performed style transfer in the frequency domain.

## 4.2 EQUIVALENCE, INTERPRETATIONS AND MANIPULATIONS

**Validation of equivalence**     For AdaIN and WCT, we implement them in the frequency domain based on Eq. (6), shown from the $3^{rd}$ to the $6^{th}$ row in Figure 1. It can be observed that these two implemented UST algorithms in the frequency domain produce the same visual effect with original algorithms. This observation validates the proposed equivalence.

**Interpretations on amplitude and phase**     To validate the roles of amplitude and phase, we replace the Fourier amplitude or phase of stylized feature maps in each layer during the stylization and present the results in Figure 2. It can be observed that feature maps with the same Fourier phase produce images with highly similar spatial arrangements. This observation matches up with our interpretations on phase, provided by its connection with the content loss in Eq. (7). On the other hand, feature maps with same Fourier amplitude produces images with highly similar contrast and intensities in colors. This observation aligns with our interpretations on amplitude, supported by its connection with Gram matrices in Eq. (25).

**Stylization manipulations**     First, we empirically display the effect of PR in Section 3.3 for image stylization, whose results are shown in Figure 3. It can be observed that for results without PR, the

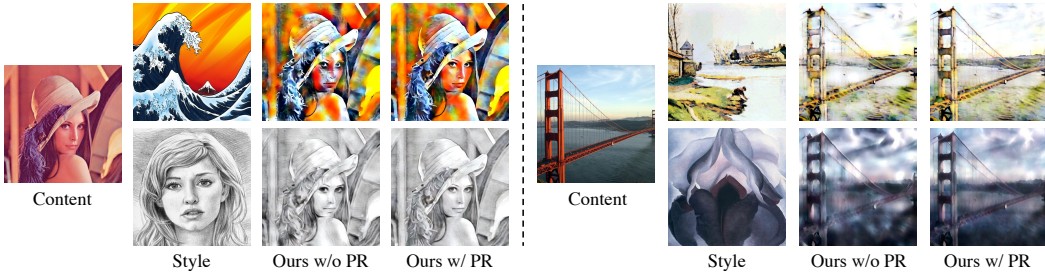

Content
Style   Ours w/o PR   Ours w/ PR
Content
Style   Ours w/o PR   Ours w/ PR

Figure 3: Results on the efficacy of phase replacement (*abbr.* PR).

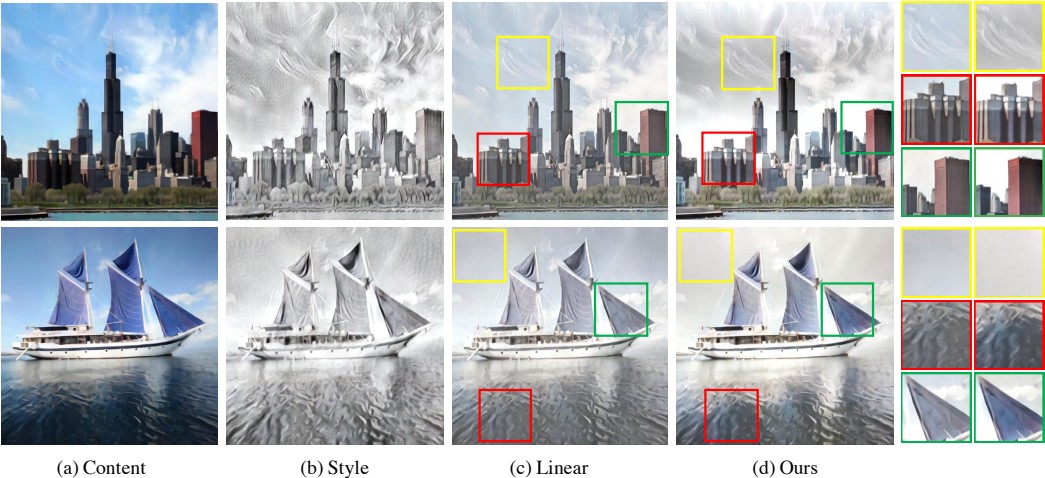

(a) Content          (b) Style          (c) Linear          (d) Ours

Figure 4: Comparison between (c) linear combination and (d) frequency combination. It is worth noting that the proposed method helps the background remain stylized (e.g., the sky more sketch-stylized than (c)) while renders the details more realistic (e.g., the buildings and the sailboat more realistic than (c)). Note that the proposed method achieves this performance without any spatial masks to identify where to stylize. The hyper-parameter $\sigma$ for results in (d) is set to 0.99.

details (*e.g.*, edges of the eyes and the nose) and the overview (*e.g.*, the sky and the sea) become messier and more distorted, yielding unappealing distortions. The reason might be that PR can preserve the phase of both high frequencies and low frequencies, which are responsible for the spatial arrangement of the details and overview of images, respectively.

Second, to demonstrate the manipulations of frequency combination (FC) in Section 3.3, we present an example in Figure 4. We choose the weighting function $\alpha$ in Eq. (12) and adjust the hyper-parameter $\sigma$ for stylization controls. In Figure 4, with different value of $\sigma$, FC can have the details less stylized (*e.g.*, the colorful buildings in the $4^{th}$ column) while keeping the background moderately stylized (*e.g.*, the sky with sketch style in the $4^{th}$ column). Furthermore, FC can be customized for various purposes and the linear combination in Eq. (10) can be viewed as an instance of FC by setting the weighting function $\alpha(u, v)$ as a simple scalar. Therefore, the controllability of FC is better than that of linear combination.

## 5 CONCLUSION

In this paper, we apply Fourier analysis to a unified framework of UST algorithms. We present the equivalent form of the framework and reveal the connections between the concepts of Fourier transform with those of style transfer. We give interpretations on the different performances between UST methods in structure preservation. We also present two operations for structure preservation and desired stylization. Extensive experiments are conducted to demonstrate (1) the equivalence between the framework and its proposed form, (2) the interpretability prompted by Fourier analysis upon style transfer and (3) the controllability through manipulations on frequency components.

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

## A    APPENDIX

In Supplementary Materials, we first provide the proofs of the main conclusions in Section 3, then we introduce the model architectures, experimental details and computing infrastructure, and finally we show the additional experimental results on our method for the performance comparison and the validation of interpretability and controllability.

## B    PROOFS

### B.1    PROOFS FOR THE EQUIVALENT FORM OF THE FRAMEWORK IN SECTION 3.1

Given the content image $I^c$ and the style image $I^s$, we denote the feature maps of $I^c$ as $F^c \in \mathbb{R}^{C \times H^c \times W^c}$ and the feature maps of $I^s$ as $F^s \in \mathbb{R}^{C \times H^s \times W^s}$, where $C$ is the number of channels,

$H^c$ ($H^s$) is the height and $W^c$ ($W^s$) is the width. Universal style transfer (UST) algorithms transform $F^c$ into the stylized feature maps $F^{cs}$ and synthesize stylized images. Based on the unified framework (Lu et al., 2019), each pixel of $F^{cs}$ is calculated as:

$$F_{h,w}^{cs} = T\left(F_{h,w}^c - \mu^c\right) + \mu^s, \tag{13}$$

where $T \in \mathbb{R}^{C \times C}$ is a transformation matrix, $(h, w)$ are indices on the spatial dimensions and $\mu^c, \mu^s \in \mathbb{R}^C$ are the channel-wise mean for $F^c$ and $F^s$, respectively. We denote the 2-D discrete Fourier transform (DFT) of $F^{cs}$ as $\mathcal{F}^{cs} \in \mathbb{C}^{C \times H^c \times W^c}$, where $\mathbb{C}$ is the set of complex numbers. Based on the Fourier transform and Eq. equation 13, we derive each complex frequency component $\mathcal{F}_{u,v}^{cs}$ as:

$$\mathcal{F}_{u,v}^{cs} = \sum_{h=0}^{H^c-1} \sum_{w=0}^{W^c-1} F_{h,w}^{cs} e^{-j2\pi(u\frac{h}{H^c}+v\frac{w}{W^c})} = \sum_{h=0}^{H^c-1} \sum_{w=0}^{W^c-1} [T(F_{h,w}^c - \mu^c) + \mu^s] e^{-j2\pi(u\frac{h}{H^c}+v\frac{w}{W^c})}$$

$$= T\underbrace{\sum_{h=0}^{H^c-1} \sum_{w=0}^{W^c-1} F_{h,w}^c e^{-j2\pi(u\frac{h}{H^c}+v\frac{w}{W^c})}}_{\text{the first term}} + \underbrace{(\mu^s - T\mu^c) \sum_{h=0}^{H^c-1} \sum_{w=0}^{W^c-1} e^{-j2\pi(u\frac{h}{H^c}+v\frac{w}{W^c})}}_{\text{the second term}},$$

$$\tag{14}$$

where $(u, v)$ are indices on the frequency dimensions and $j$ is the imaginary unit. Note that the first term of $\mathcal{F}_{u,v}^{cs}$ equals:

$$T\sum_{h=0}^{H^c-1} \sum_{w=0}^{W^c-1} F_{h,w}^c e^{-j2\pi(u\frac{h}{H}+v\frac{w}{W})} = T\mathcal{F}_{u,v}^c. \tag{15}$$

Especially, when $u = v = 0$, $T\mathcal{F}_{u,v}^c$ equals:

$$T\mathcal{F}_{0,0}^c = T\sum_{h=0}^{H^c-1} \sum_{w=0}^{W^c-1} F_{h,w}^c e^{0+0} = T(H^c W^c \mu^c). \tag{16}$$

And the second term of $\mathcal{F}_{u,v}^{cs}$ equals:

$$(\mu^s - T\mu^c) \sum_{h=0}^{H^c-1} \sum_{w=0}^{W^c-1} e^{-j2\pi(u\frac{h}{H}+v\frac{w}{W})} = \begin{cases} H^c W^c (\mu^s - T\mu^c), & \text{if } u = v = 0; \\ 0, & \text{else} \end{cases}. \tag{17}$$

Thus, $\mathcal{F}_{u,v}^{cs}$ can be derived as:

$$\mathcal{F}_{u,v}^{cs} = \begin{cases} T(H^c W^c \mu^c) - T(H^c W^c \mu^c) + H^c W^c \mu^s, & \text{if } u = v = 0; \\ T\mathcal{F}_{u,v}^c, & \text{else} \end{cases}$$

$$= \begin{cases} H^c W^c \mu^s, & \text{if } u = v = 0; \\ T\mathcal{F}_{u,v}^c, & \text{else} \end{cases} = \begin{cases} \left(\frac{H^c W^c}{H^s W^s}\right) \mathcal{F}_{0,0}^s, & \text{if } u = v = 0; \\ T\mathcal{F}_{u,v}^c, & \text{else} \end{cases}, \tag{18}$$

where $\mathcal{F}^c$ and $\mathcal{F}^s$ are the DFTs of $F^c$ and $F^s$, respectively. It is worth noting that $\mathcal{F}_{0,0}^s = \sum_{h=0}^{H^s-1} \sum_{w=0}^{W^s-1} F_{h,w}^s = H^s W^s \mu^s$. Therefore, we have $\mathcal{F}_{u,v}^{cs} = H^c W^c \mu^s = \left(\frac{H^c W^c}{H^s W^s}\right) \mathcal{F}_{0,0}^s$ when $u = v = 0$.

### B.2 PROOFS FOR THE CONNECTION BETWEEN PHASE AND THE CONTENT LOSS IN SECTION 3.2

Given matrices $X, Y \in \mathbb{C}^{H \times W}$ and their 2-D DFTs $\mathcal{X}, \mathcal{Y} \in \mathbb{C}^{H \times W}$, we have the Parseval's theorem in the context of DFT that:

$$\sum_{h=0}^{H-1} \sum_{w=0}^{W-1} X_{h,w} Y_{h,w}^* = \frac{1}{HW} \sum_{u=0}^{H-1} \sum_{v=0}^{W-1} \mathcal{X}_{u,v} \mathcal{Y}_{u,v}^*, \tag{19}$$

where $(*)$ represents complex conjugate and $u, v$ are indices on the frequency dimensions. Especially, when $X = Y$ and $X, Y \in \mathbb{R}^{H \times W}$, we can have:

$$\sum_{h=0}^{H-1} \sum_{w=0}^{W-1} (X_{h,w})^2 = \frac{1}{HW} \sum_{u=0}^{H-1} \sum_{v=0}^{W-1} |\mathcal{X}_{u,v}|^2. \tag{20}$$

We then denote the feature maps of the stylized image and the content image as $F^{cs}, F^{c} \in \mathbb{R}^{C \times H \times W}$, their corresponding DFTs as $\mathcal{F}^{cs}, \mathcal{F}^{c}$, Fourier amplitude as $|\mathcal{F}^{cs}|, |\mathcal{F}^{c}|$ and Fourier phase as $\angle \mathcal{F}^{cs}, \angle \mathcal{F}^{c}$, respectively. Note that for any $\mathcal{F}^{cs}_{k,u,v}$ and $\mathcal{F}^{c}_{k,u,v}$, we have:

$$
\begin{aligned}
\mathcal{F}^{cs}_{k,u,v} &= |\mathcal{F}^{cs}_{k,u,v}| \cos \angle \mathcal{F}^{cs}_{k,u,v} + j|\mathcal{F}^{cs}_{k,u,v}| \sin \angle \mathcal{F}^{cs}_{k,u,v} \\
\mathcal{F}^{c}_{k,u,v} &= |\mathcal{F}^{c}_{k,u,v}| \cos \angle \mathcal{F}^{c}_{k,u,v} + j|\mathcal{F}^{c}_{k,u,v}| \sin \angle \mathcal{F}^{c}_{k,u,v},
\end{aligned}
\tag{21}
$$

where $k$ and $(u,v)$ are indices on channels and frequency dimensions, respectively. Note that for complex numbers $\mathcal{M} = a + jb$ and $\mathcal{N} = c + jd$ with $a, b, c, d \in \mathbb{R}$, we have:

$$
|\mathcal{M} - \mathcal{N}|^2 = (a - c)^2 + (b - d)^2. \tag{22}
$$

Based on the Eq. equation 20, Eq. equation 21 and Eq. equation 22, the content loss $\mathcal{L}_c$ between $F^{cs}$ and $F^{c}$ can be represented as:

$$
\begin{aligned}
\mathcal{L}_c &= \sum_{k=0}^{C-1} \sum_{h=0}^{H-1} \sum_{w=0}^{W-1} \left( F^{cs}_{k,h,w} - F^{c}_{k,h,w} \right)^2 = \frac{1}{HW} \sum_{k=0}^{C-1} \sum_{u=0}^{H-1} \sum_{v=0}^{W-1} |\mathcal{F}^{cs}_{k,u,v} - \mathcal{F}^{c}_{k,u,v}|^2 \\
&= \frac{1}{HW} \sum_{k=0}^{C-1} \sum_{u=0}^{H-1} \sum_{v=0}^{W-1} \Big[ \left( |\mathcal{F}^{cs}_{k,u,v}| \cos \angle \mathcal{F}^{cs}_{k,u,v} - |\mathcal{F}^{c}_{k,u,v}| \cos \angle \mathcal{F}^{c}_{k,u,v} \right)^2 \\
&\qquad\qquad\qquad\qquad\qquad + \left( |\mathcal{F}^{cs}_{k,u,v}| \sin \angle \mathcal{F}^{cs}_{k,u,v} - |\mathcal{F}^{c}_{k,u,v}| \sin \angle \mathcal{F}^{c}_{k,u,v} \right)^2 \Big] \\
&= \frac{1}{HW} \sum_{k=0}^{C-1} \sum_{u=0}^{H-1} \sum_{v=0}^{W-1} \Big( |\mathcal{F}^{cs}_{k,u,v}|^2 + |\mathcal{F}^{c}_{k,u,v}|^2 - 2|\mathcal{F}^{cs}_{k,u,v}||\mathcal{F}^{c}_{k,u,v}| \cos \angle \mathcal{F}^{cs}_{k,u,v} \cos \angle \mathcal{F}^{c}_{k,u,v} \\
&\qquad\qquad\qquad\qquad\qquad - 2|\mathcal{F}^{cs}_{k,u,v}||\mathcal{F}^{c}_{k,u,v}| \sin \angle \mathcal{F}^{cs}_{k,u,v} \sin \angle \mathcal{F}^{c}_{k,u,v} \Big) \\
&= \frac{1}{HW} \sum_{k=0}^{C-1} \sum_{u=0}^{H-1} \sum_{v=0}^{W-1} \Big[ |\mathcal{F}^{cs}_{k,u,v}|^2 + |\mathcal{F}^{c}_{k,u,v}|^2 - 2|\mathcal{F}^{cs}_{k,u,v}||\mathcal{F}^{c}_{k,u,v}| \cos \left( \angle \mathcal{F}^{cs}_{k,u,v} - \angle \mathcal{F}^{c}_{k,u,v} \right) \Big].
\end{aligned}
\tag{23}
$$

It is worth noting that $\cos(\alpha - \beta) = \cos \alpha \cos \beta + \sin \alpha \sin \beta$ for any $\alpha, \beta \in (-\pi, \pi]$.

### B.3 PROOFS FOR THE CONNECTION BETWEEN AMPLITUDE AND GRAM MATRICES IN SECTION 3.2

According to Eq. equation 19, when $X, Y \in \mathbb{R}^{H \times W}$, $Y_{h,w} = Y^*_{h,w}$, we have:

$$
\sum_{h=0}^{H-1} \sum_{w=0}^{W-1} X_{h,w} Y_{h,w} = \frac{1}{HW} \sum_{u=0}^{H-1} \sum_{v=0}^{W-1} \mathcal{X}_{u,v} \mathcal{Y}^*_{u,v}, \tag{24}
$$

which serves as the basis for our following proofs. Given feature maps $F \in \mathbb{R}^{C \times H \times W}$, their corresponding DFTs $\mathcal{F}$, Fourier amplitude $|\mathcal{F}|$ and Fourier phase $\angle \mathcal{F}$, the pixel of the Gram matrix $FF^\top$ can be derived as:

$$
\begin{aligned}
\left( FF^\top \right)_{c_1, c_2} &= \sum_{h=0}^{H-1} \sum_{w=0}^{W-1} F_{c_1, h, w} F_{c_2, h, w} = \frac{1}{HW} \sum_{u=0}^{H-1} \sum_{v=0}^{W-1} \mathcal{F}_{c_1, u, v} \mathcal{F}^*_{c_2, u, v} \\
&= \frac{1}{HW} \sum_{u=0}^{H-1} \sum_{v=0}^{W-1} \Big[ |\mathcal{F}_{c_1, u, v}||\mathcal{F}_{c_2, u, v}| \cos \left( \angle \mathcal{F}_{c_1, u, v} - \angle \mathcal{F}_{c_2, u, v} \right) \\
&\qquad\qquad\qquad\qquad + j|\mathcal{F}_{c_1, u, v}||\mathcal{F}_{c_2, u, v}| \sin \left( \angle \mathcal{F}_{c_1, u, v} - \angle \mathcal{F}_{c_2, u, v} \right) \Big] \\
&= \frac{1}{HW} \sum_{u=0}^{H-1} \sum_{v=0}^{W-1} |\mathcal{F}_{c_1, u, v}||\mathcal{F}_{c_2, u, v}| \cos \left( \angle \mathcal{F}_{c_1, u, v} - \angle \mathcal{F}_{c_2, u, v} \right),
\end{aligned}
\tag{25}
$$

where $c_1, c_2$ are indices on the channels. Considering that $\left( FF^\top \right)_{c_1, c_2}$ is a real number, the imaginary part $\frac{1}{HW} \sum_{u=0}^{H-1} \sum_{v=0}^{W-1} j|\mathcal{F}_{c_1, u, v}||\mathcal{F}_{c_2, u, v}| \sin \left( \angle \mathcal{F}_{c_1, u, v} - \angle \mathcal{F}_{c_2, u, v} \right)$ equals zero and is consequently omitted in the final step.

## C  IMPLEMENTATION DETAILS

### C.1  METHOD SPECIFICATION

The encoder of the proposed method is a part of the VGG-19 network (Simonyan & Zisserman, 2015) (from the layer *conv*1_1 to the layer *conv*4_1). The decoder is a generative network to perform the stylization and transform the stylized feature maps into images. The decoder performs both *whitening and coloring transform* (WCT) (Li et al., 2017a) and the proposed phase replacement operation in the frequency domain only for the inference stage. The detailed network architecture is described as follows:

- The encoder:
  - $1 \times 1$ Conv, 3
  - $1 \times 1$ Reflection Padding, $3 \times 3$ Conv, 64 ReLU
  - $1 \times 1$ Reflection Padding, $3 \times 3$ Conv, 64 ReLU
  - $2 \times 2$ Max pool, stride 2; $1 \times 1$ Reflection Padding, $3 \times 3$ Conv, 128 ReLU
  - $1 \times 1$ Reflection Padding, $3 \times 3$ Conv, 128 ReLU
  - $2 \times 2$ Max pool, stride 2; $1 \times 1$ Reflection Padding, $3 \times 3$ Conv, 256 ReLU
  - $1 \times 1$ Reflection Padding, $3 \times 3$ Conv, 256 ReLU
  - $1 \times 1$ Reflection Padding, $3 \times 3$ Conv, 256 ReLU
  - $1 \times 1$ Reflection Padding, $3 \times 3$ Conv, 256 ReLU
  - $2 \times 2$ Max pool, stride 2; $1 \times 1$ Reflection Padding, $3 \times 3$ Conv, 512 ReLU

- The decoder:
  - Universal style transfer module; Phase replacement
  - $1 \times 1$ Reflection Padding, $3 \times 3$ Conv, 256 ReLU
  - 2x nearest-neighbor upsample; $1 \times 1$ Reflection Padding, $3 \times 3$ Conv, 256 ReLU
  - $1 \times 1$ Reflection Padding, $3 \times 3$ Conv, 256 ReLU
  - $1 \times 1$ Reflection Padding, $3 \times 3$ Conv, 256 ReLU
  - Universal style transfer module; Phase replacement
  - $1 \times 1$ Reflection Padding, $3 \times 3$ Conv, 128 ReLU
  - 2x nearest-neighbor upsample; $1 \times 1$ Reflection Padding, $3 \times 3$ Conv, 128 ReLU
  - Universal style transfer module; Phase replacement
  - $1 \times 1$ Reflection Padding, $3 \times 3$ Conv, 64 ReLU
  - 2x nearest-neighbor upsample; $1 \times 1$ Reflection Padding, $3 \times 3$ Conv, 64 ReLU
  - Universal style transfer module; Phase replacement
  - $1 \times 1$ Reflection Padding, $3 \times 3$ Conv, 3

### C.2  EXPERIMENTAL DETAILS

The encoder of the proposed method is pre-trained by the ImageNet dataset (**?**). The decoder of the proposed method is trained for image reconstruction by minimizing the $L_2$ reconstruction loss. The corresponding training dataset is MS-COCO dataset (Lin et al., 2014), whose images are first resized into $512 \times 512$ and randomly cropped into a size of $256 \times 256$ for training. During the learning stage, the batch size is 64 and the number of training steps is 140,000. The initial learning rate is $1 \times 10^{-4}$, and is decayed with a factor 0.5 for every 20,000 steps. For the stylization, we choose MS-COCO dataset (Lin et al., 2014) and WikiArt dataset (Nichol, 2016) as our content dataset and style dataset, respectively. For the quantitative comparison, we randomly select 15 content images and 25 style images to synthesize 375 stylized images for each method.

### C.3  COMPUTING INFRASTRUCTURE

Experiments are conducted on a server with Intel Xeon Silver 4214 CPUs and a single NVIDIA Tesla V100 GPU. The code is developed based on the PyTorch framework (**?**) with version 1.8.1.

Table 2: The SSIM scores for different methods on the architectures of AdaIN and WCT.

| Method \ Architecture | AdaIN | WCT |
|---|---|---|
| AdaIN | **0.307** | **0.315** |
| WCT | 0.274 | 0.234 |
| WCT w/ PR | 0.280 | 0.251 |

## D  INVOLVED EXISTING ASSETS

Existing assets involved in this work include: 1) the codes and models weights of AdaIN (Huang & Belongie, 2017), WCT (Li et al., 2017a), LinearWCT (Li et al., 2019), Avatar-Net (Sheng et al., 2018), OptimalWCT (Lu et al., 2019), SANet (Park & Lee, 2019), and Self-Contained (Chen et al., 2020), and 2) the images of MS-COCO dataset (Lin et al., 2014) and WikiArt dataset (Nichol, 2016). Their URLs and licenses are as follows:

- AdaIN: https://github.com/naoto0804/pytorch-AdaIN. MIT License.
- WCT: https://github.com/irasin/Pytorch_WCT. Unable to find its license.
- LinearWCT: https://github.com/sunshineatnoon/LinearStyleTransfer. BSD 2-Clause License.
- Avatar-Net: https://github.com/LucasSheng/avatar-net. Unable to find its license.
- OptimalWCT: https://github.com/boomb0om/PyTorch-OptimalStyleTransfer. Unable to find its license.
- MS-COCO dataset: https://cocodataset.org/#download. Unable to find its license.
- WikiArt dataset: https://www.kaggle.com/c/painter-by-numbers. Unable to find its license.
- SANet: https://github.com/GlebSBrykin/SANET
- Self-Contained: https://github.com/IShengFang/Self-Contained_Stylization

## E  ADDITIONAL EXPERIMENTAL RESULTS

In this section, we present more experimental results in this work, including (1) more performance comparison against the state-of-the-art style transfer methods in Figure 6, (2) more validation on the interpretations of Fourier phase and amplitude in Figure 7 and Figure 8, and (3) more experiments on the efficacy of proposed manipulations in Table 2, Figure 9 and Figure 10.

Note that the structure preservation abilities of different methods are not only related with their stylization transformations on feature maps, but also related with their architectures and model weights. To better understand the differences between AdaIN and WCT, we implement different methods on their architectures and present corresponding quantitative scores. As shown in Table 2, WCT is improved by phase replacement (PR) while AdaIN achieves the highest SSIM scores. The reason might be that WCT has more influence on the Fourier amplitude of feature maps compared with AdaIN. According to Eq. equation 23, when the Fourier phase is well preserved by AdaIN and PR, the content loss only depends on the differences of Fourier amplitude, which means that changing Fourier amplitude is likely to harm the SSIM score. Since WCT has lower SSIM scores compared with AdaIN, it is more likely to influence more on the Fourier amplitude. This also explains why WCT can produce much more intensive and expressive stylization, considering the connection between Fourier amplitude and the intensities of stylized images.

## REFERENCES

Prashanth Chandran, Gaspard Zoss, Paulo Gotardo, Markus Gross, and Derek Bradley. Adaptive convolutions for structure-aware style transfer. In *Proceedings of the Conference on Computer Vision and Pattern Recognition (CVPR)*, pp. 7972–7981, June 2021.

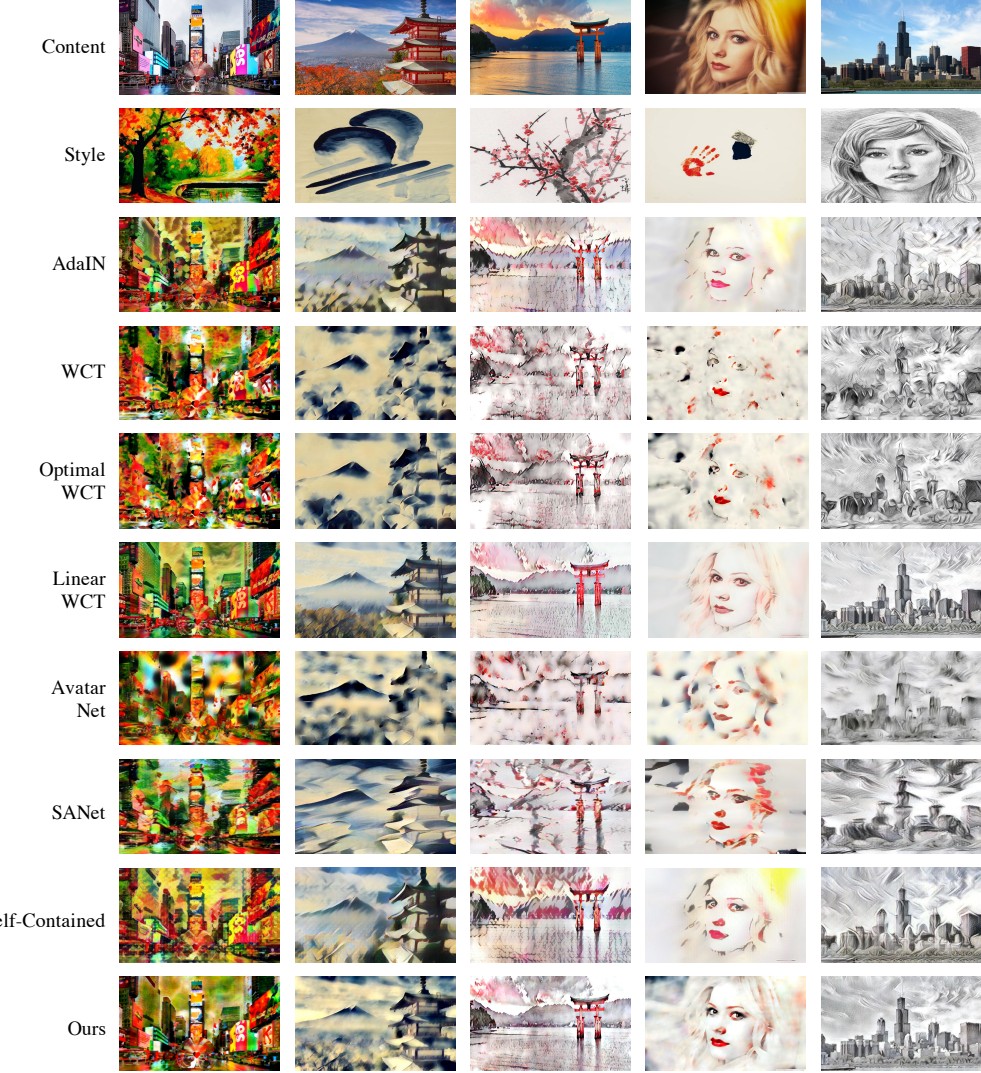

Figure 5: Qualitative comparison on the state-of-the-art UST algorithms. Compared with the proposed method, AvatarNet and SANet sometimes generates less intensive colors (*e.g.*, in $2^{nd}$ and $4^{th}$ columns) and produce unexpected style pattern (*e.g.*, eyes in the $5^{th}$ column), while Self-Contained sometimes introduces unexpected color (*e.g.*, in the $4^{th}$ column)

Haibo Chen, lei zhao, Zhizhong Wang, Huiming Zhang, Zhiwen Zuo, Wei Xing Ailin Li, and Dongming Lu. Artistic style transfer with internal-external learning and contrastive learning. In M. Ranzato, A. Beygelzimer, Y. Dauphin, P.S. Liang, and J. Wortman Vaughan (eds.), *Advances in Neural Information Processing Systems*, volume 34, pp. 26561–26573. Curran Associates, Inc., 2021a. URL https://proceedings.neurips.cc/paper/2021/file/df5354693177e83e8ba089e94b7b6b55-Paper.pdf.

Haibo Chen, Lei Zhao, Zhizhong Wang, Huiming Zhang, Zhiwen Zuo, Ailin Li, Wei Xing, and Dongming Lu. Dualast: Dual style-learning networks for artistic style transfer. In *Proceedings of the Conference on Computer Vision and Pattern Recognition (CVPR)*, pp. 872–881, June 2021b.

Haibo Chen, Lei Zhao, Huiming Zhang, Zhizhong Wang, Zhiwen Zuo, Ailin Li, Wei Xing, and Dongming Lu. Diverse image style transfer via invertible cross-space mapping. In *Proceedings of the International Conference on Computer Vision (ICCV)*, pp. 14880–14889, October 2021c.

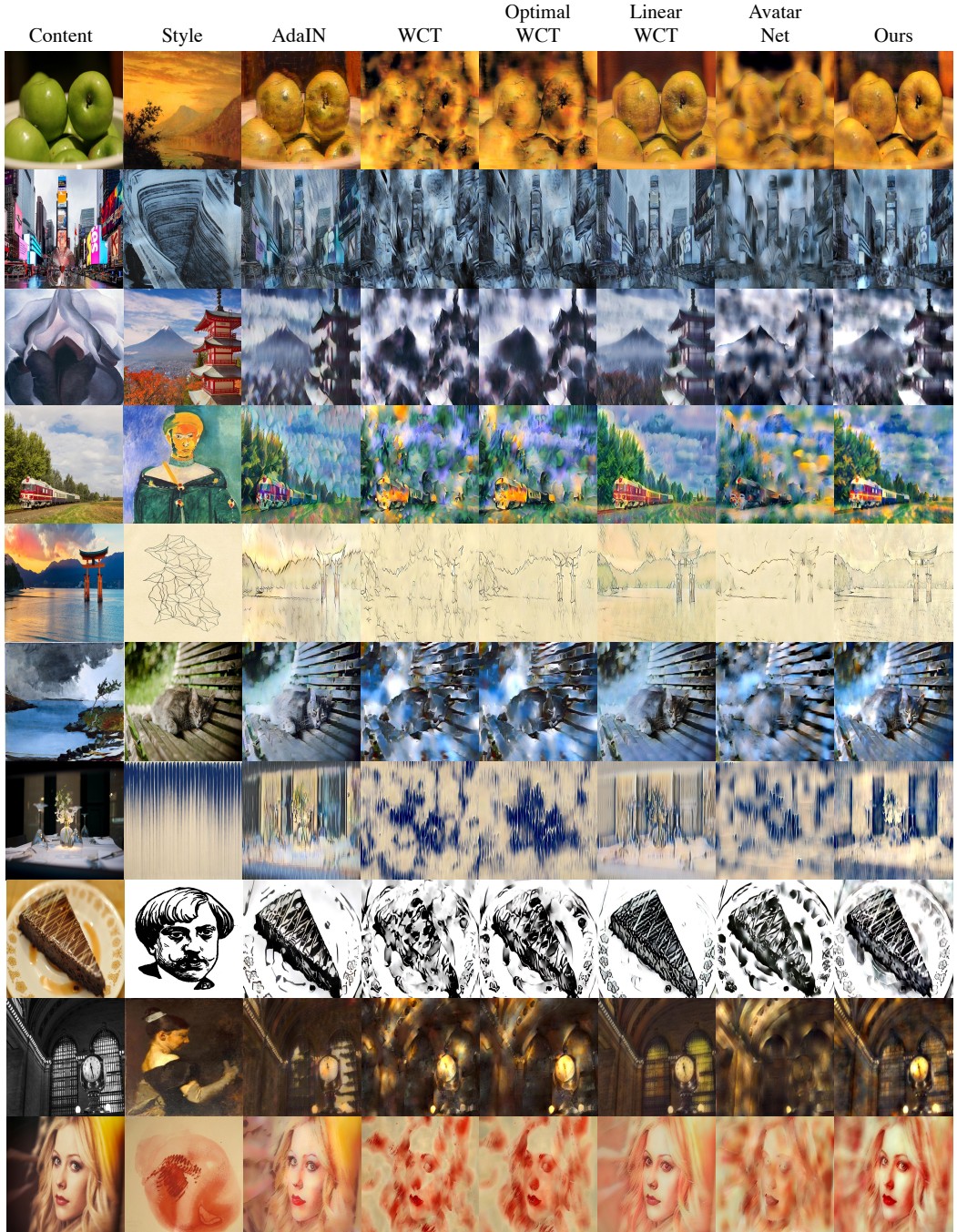

Figure 6: Qualitative comparison on the state-of-the-art UST algorithms. On one hand, compared with AdaIN and LinearWCT, the proposed method produces more intensive colors (*e.g.*, $3^{rd}$, $4^{th}$, $6^{th}$ and $9^{th}$ rows). On the other hand, the proposed method does better in the structure preservation of images than the rest of methods (*e.g.*, $1^{st}$, $2^{nd}$, $5^{th}$, $7^{th}$ and $8^{th}$ rows).

Hung-Yu Chen, I-Sheng Fang, Chia-Ming Cheng, and Wei-Chen Chiu. Self-contained stylization via steganography for reverse and serial style transfer. In *Proceedings of the IEEE/CVF Winter Conference on Applications of Computer Vision (WACV)*, March 2020.

Tai-Yin Chiu and Danna Gurari. Photowct 2 : Compact autoencoder for photorealistic style transfer resulting from blockwise training and skip connections of high-frequency residuals. pp. 2978–

| (a) Content image | (b) Style image | (c) Stylized image | (d) Amplitude of (b) and phase of (c) | (e) Amplitude of (a) and phase of (c) |

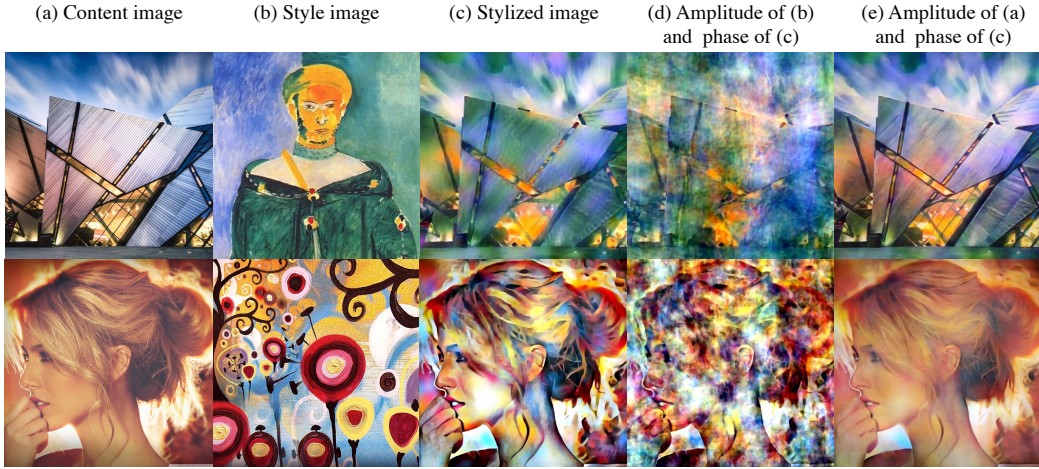

Figure 7: Additional synthesized results with the same Fourier phase and replaced Fourier amplitude. We replace the Fourier amplitude of stylized feature maps in each layer and use the proposed method to synthesize stylized images. Note that feature maps with the same phase produce images with highly similar spatial arrangements (*e.g.*, the triangular structure of the architecture and the human face), even in some details (*e.g.*, the light on the human cheek). Therefore, Fourier phase is validated to play an important role on the spatial arrangements of images.

| (a) Content image | (b) Style image | (c) Stylized image | (d) Phase of (b) and amplitude of (c) | (e) Phase of (a) and amplitude of (c) |

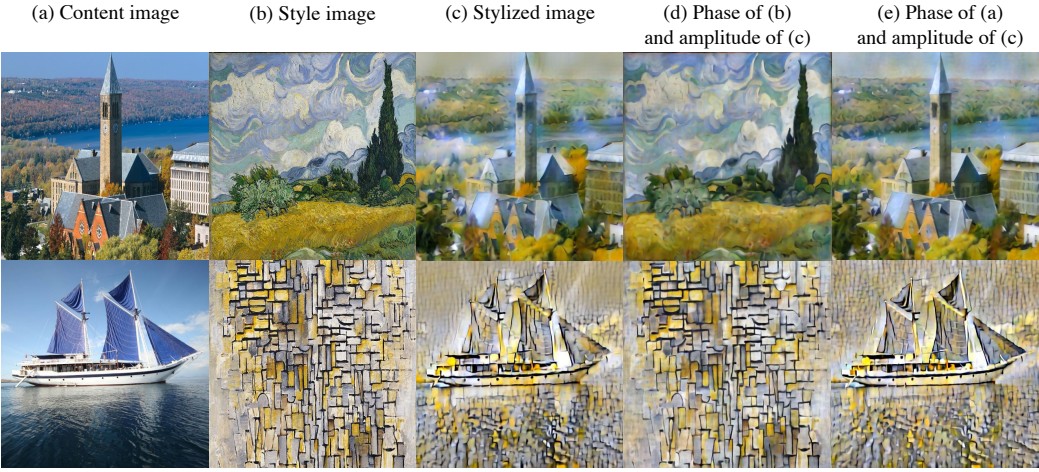

Figure 8: Additional synthesized results with the same Fourier amplitude and replaced Fourier phase. We replace the Fourier phase of stylized feature maps in each layer and use the proposed method (without phase replacement) to synthesize stylized images. Note that feature maps with the same amplitude produce images with highly similar contrast and intensities, (*e.g.*, the same differences between the brightest pixels and the darkest pixels of (c), (d) and (e)). Therefore, we validate the connection between Fourier amplitude and the intensities of synthesized images.

2987, 01 2022. doi: 10.1109/WACV51458.2022.00303.

Vincent Dumoulin, Jonathon Shlens, and Manjunath Kudlur. Learned representation for artistic style. In *International Conference on Learning Representations*, 2017.

Leon A. Gatys, Alexander S. Ecker, and Matthias Bethge. Image style transfer using convolutional neural networks. In *Proceedings of the IEEE Conference on Computer Vision and Pattern Recognition (CVPR)*, pp. 2414–2423, 2016. doi: 10.1109/CVPR.2016.265.

Rafael C. Gonzalez and Richard E. Woods. *Digital image processing*, pp. 286–306. Prentice Hall, Upper Saddle River, N.J., 2008. ISBN 9780131687288

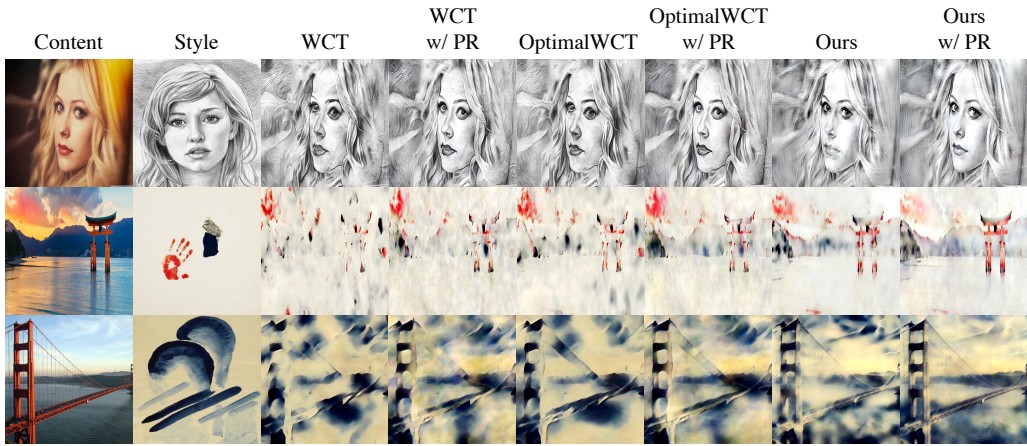

Figure 9: Results on the efficacy of phase replacement (*abbr.* PR). We implement PR on different methods and show the visual difference. It can be observed that PR improves the visual effect of all methods in the spatial arrangement of not only the details (*e.g.*, it improves the unpleasant human eyes and vague contours of architectures ), but also the overview (*e.g.*, it improves the uneven light and shade) of images.

013168728X 9780135052679 013505267X. URL `http://www.amazon.com/Digital-Image-Processing-3rd-Edition/dp/013168728X`.

Kibeom Hong, Seogkyu Jeon, Huan Yang, Jianlong Fu, and Hyeran Byun. Domain-aware universal style transfer. In *Proceedings of the International Conference on Computer Vision (ICCV)*, pp. 14609–14617, October 2021.

Xun Huang and Serge Belongie. Arbitrary style transfer in real-time with adaptive instance normalization. In *International Conference on Computer Vision (ICCV)*, pp. 1510–1519, 2017. doi: 10.1109/ICCV.2017.167.

William Frost Jenkins and Mita D. Desai. The discrete frequency fourier transform. *IEEE Transactions on Circuits and Systems*, 33:732–734, 1986.

Justin Johnson, Alexandre Alahi, and Li Fei-Fei. European conference on computer vision. In *Perceptual losses for real-time style transfer and super-resolution*, pp. 694–711, 2016.

Chuan Li and Michael Wand. Combining markov random fields and convolutional neural networks for image synthesis. In *Proceedings of the IEEE Conference on Computer Vision and Pattern Recognition*, pp. 2479–2486, 2016.

Xueting Li, Sifei Liu, Jan Kautz, and Ming-Hsuan Yang. Learning linear transformations for fast image and video style transfer. In *Conference on Computer Vision and Pattern Recognition (CVPR)*, pp. 3804–3812, 2019. doi: 10.1109/CVPR.2019.00393.

Yijun Li, Chen Fang, Jimei Yang, Zhaowen Wang, Xin Lu, and Ming-Hsuan Yang. Universal style transfer via feature transforms. In *Advances in neural information processing systems*, pp. 386–396, 2017a.

Yijun Li, Chen Fang, Jimei Yang, Zhaowen Wang, Xin Lu, and Ming-Hsuan Yang. Diversified texture synthesis with feed-forward networks. In *IEEE Conference on Computer Vision and Pattern Recognition (CVPR)*, 2017b.

Yijun Li, Ming-Yu Liu, Xueting Li, Ming-Hsuan Yang, and Jan Kautz. A closed-form solution to photorealistic image stylization. In *Proceedings of the European Conference on Computer Vision (ECCV)*, September 2018.

Tsung-Yi Lin, Michael Maire, Serge Belongie, Lubomir Bourdev, Ross Girshick, James Hays, Pietro Perona, Deva Ramanan, C. Lawrence Zitnick, and Piotr Dollár. Microsoft coco: Common objects in context, 2014. URL `https://arxiv.org/abs/1405.0312`.

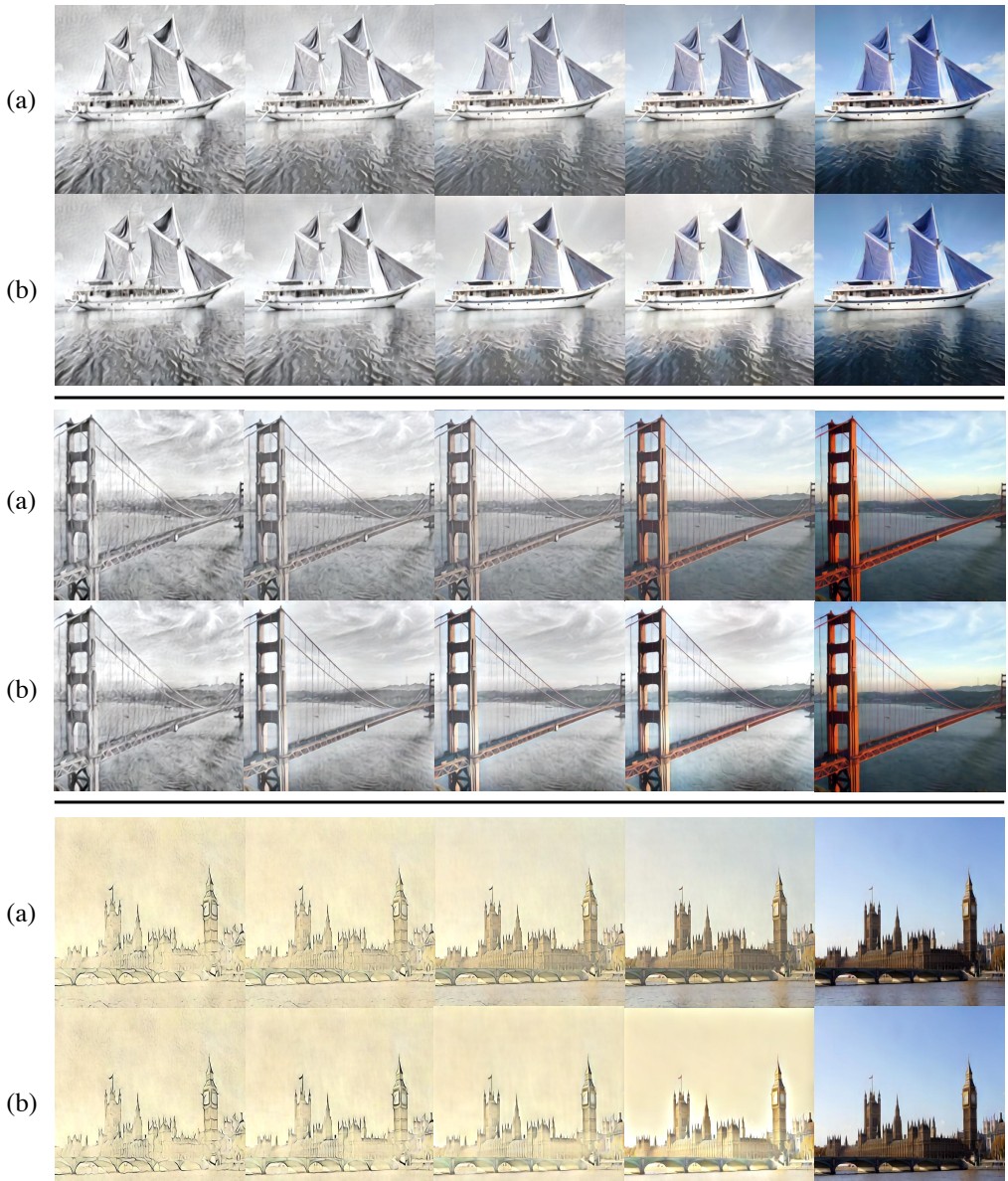

Figure 10: Comparison between (a) linear combination and (b) frequency combination. Images in the first column are stylized images and images in the last column are content images. The proposed frequency method can maintain the low frequencies of stylized images and focus on the high frequencies of content images. On one hand, the proposed method can fix distorted details by introducing the high frequencies of content images, since these distortion are caused by the high frequencies of stylized images (*e.g.*, the difference between the first column and the second column of (b)). On the other hand, the proposed method can enhance the details while keeping the overview stylized (*e.g.*, the difference between the fourth column of (a) and (b)). Note that the proposed method achieves this performance without any spatial masks to identify where to stylize.

Xiao-Chang Liu, Yong-Liang Yang, and Peter Hall. Learning to warp for style transfer. In *Proceedings of the Conference on Computer Vision and Pattern Recognition (CVPR)*, pp. 3702–3711, June 2021.

Ming Lu, Hao Zhao, Anbang Yao, Yurong Chen, Feng Xu, and Li Zhang. A closed-form solution to universal style transfer. In *International Conference on Computer Vision (ICCV)*, pp. 5951–5960,

2019.

K . Nichol. Painter by numbers. volume 34, 2016. URL `https://www.kaggle.com/c/painter-by-numbers`.

Dae Young Park and Kwang Hee Lee. Arbitrary style transfer with style-attentional networks. In *Proceedings of the IEEE International Conference on Computer Vision*, pp. 5880–5888, 2019.

Lu Sheng, Ziyi Lin, Jing Shao, and Xiaogang Wang. Avatar-net: Multi-scale zero-shot style transfer by feature decoration, 2018. URL `https://arxiv.org/abs/1805.03857`.

Karen Simonyan and Andrew Zisserman. Very deep convolutional networks for large-scale image recognition. In *International Conference on Learning Representations*, 2015.

Zhizhong Wang, Lei Zhao, Haibo Chen, Lihong Qiu, Qihang Mo, Sihuan Lin, Wei Xing, and Dongming Lu. Diversified arbitrary style transfer via deep feature perturbation. In *Proceedings of the IEEE International Conference on Computer Vision*, pp. 7789–7798, 2020.

Yanchao Yang and Stefano Soatto. Fda: Fourier domain adaptation for semantic segmentation. In *Proceedings of the IEEE/CVF Conference on Computer Vision and Pattern Recognition (CVPR)*, June 2020.

Jaejun Yoo, Youngjung Uh, Sanghyuk Chun, Byeongkyu Kang, and Jung-Woo Ha. Photorealistic style transfer via wavelet transforms. In *International Conference on Computer Vision (ICCV)*, pp. 9035–9044, 2019. doi: 10.1109/ICCV.2019.00913.

