# OpenReview forum: "Style Spectroscope: Improve Interpretability and Controllability through Fourier Analysis"
_ICLR.cc/2023/Conference — Submitted to ICLR 2023_

### Official Review · Reviewer_eNXa · 2022-10-19

**Confidence:** 4
**Correctness:** 3
**Technical Novelty And Significance:** 4
**Empirical Novelty And Significance:** 3
**Recommendation:** 8

**Clarity, Quality, Novelty And Reproducibility:**

This paper provides a novel view of UST framework in frequency domains. The analysis also provides valuable applications like structure preservation and non-uniform style transfer. The method is about Fourier transformation over the existing UST framework, meaning the re-implementation is not hard.

**Strength And Weaknesses:**

**Strengths:**
+ The analysis of current universal style transfer framework in Fourier frequency domain is interesting. And based on the analysis, the authors leverage the characteristics of the frequency domain for structure preservation and non-uniform style transfer. I think the idea is interesting and insightful. The usage also proves the application value.
+ The analysis is not limited to a simple style model but a general UST framework.

**Weaknesses:**
+ Comparing to the phase part, relating amplitude to Gram matrices is less convincing. Since amplitude is only related to the diagonals of Gram matrices, which is more similar to the channel-wise second order statistics of feature maps in the style loss of AdaIN than the Gram -based style loss. The amplitude didn’t explain the key inter-channel correlations in the Gram matrices. Compared to AdaIN, WCT also uses inter-channel correlations to model image styles. Therefore, I think it will be more appropriate to use other word than Gram matrices here.

**Some small issues:**
+ Some typos. For example, ``Next, We demonstrate``,
+ In Table I, why LinearWCT has better scores in Time but the scores of AdaIN is bold?
+ Figure 4, (b) is the style transfer results rather than the style image.


**Summary Of The Paper:**

This paper proposes a systematic Fourier analysis on the universal style transfer framework. It reveals the relationship between Fourier amplitude / phase with Gram matrices / a content reconstruction loss. Based on the analysis, this paper proposes a new structure preservation method by phase swapping and perform non-uniform style transfer based on frequency.

**Summary Of The Review:**

This paper provides a novel view of UST framework in frequency domains and validate the analysis with comprehensive experiment as well as algorithms of structure preservation and non-uniform style transfer. The analysis is interesting to me and may enlighten the follow-up style transfer researches.

---

> ### Author Response · Authors · 2022-12-06
> **Thanks for your constructive feedback!**
>
> We genuinely thank the reviewer for your thoughtful comments and high opinion of our paper. We will take your advice in the theoretical analysis and revise other issues. Many thanks again.

---

### Official Review · Reviewer_pCyN · 2022-10-21

**Confidence:** 3
**Correctness:** 3
**Technical Novelty And Significance:** 2
**Empirical Novelty And Significance:** 2
**Recommendation:** 5

**Clarity, Quality, Novelty And Reproducibility:**

The layout and writing are clear. The proposed method can be easily reproduced. The novelty of the paper is limited, except for the interesting analysis in the Fourier domain.

**Strength And Weaknesses:**

Strength

+Interesting view
This paper revisits style transfer from the view of the Fourier transform. Although the Fourier has been used in low-level vision tasks such as image restoration, it has not been used in style transfer.

+New insight
The paper provides new insights into the effect of amplitude and phase of Fourier transfer on the performance of style transfer.

+Solid proof.
This work provides theoretical proof of the equivalent form of UST methods in the frequency domain.

Weakness

-Limited improvement
From the results, even using the proposed method, the performance of style transfer is still unsatisfactory. There are artifacts in the results. Thus, it is not clear the effectiveness of the proposed method.

-Insufficient experiment
1) For the style transfer tasks, more visual results should be included, especially in complex and challenging cases.
2) More results works should be included for comparison. It seems only several previous UST methods are compared. It is difficult to see the advantages of the proposed method.
3) The paper uses only SSIM as an evaluation metric. Why not use other metrics?




**Summary Of The Paper:**

This paper focuses on style transfer. Different from previous works, this work shows the equivalent form of UST methods in the frequency domain. Moreover, the proposed method revisits existing UST methods in the frequency domain, showing the effects of Fourier amplitude and phase. With these experiments, this work proposes two manipulations for better style transfer. Some experiments are conducted to show the advantages of the proposed manipulations over previous UST methods.

**Summary Of The Review:**

The paper has some novelty and experiment flaws, which are the most important factors in my rating. In addition,  the advantage of the proposed method is not much obvious. The authors can provide comments based on the points I mentioned in the weaknesses. After that, I will consider my final decision.

---

> ### Author Response · Authors · 2022-12-06
> **Thanks for your constructive feedback!**
>
> We thank the reviewer for your time and thoughtful comments. We will take your suggestions and improve our novelty and experiments, especially in presenting more results comparison and evaluation metrics.

---

> > ### Comment · Reviewer_pCyN · 2022-12-13
> > **Final Recommendation**
> >
> > Thank you for the response. After reading the response and the comments of other reviewers, I would like to keep my original recommendation.

---

### Official Review · Reviewer_niHJ · 2022-10-23

**Confidence:** 4
**Clarity, Quality, Novelty And Reproducibility:** Please see the strength and weakness …
**Correctness:** 2
**Technical Novelty And Significance:** 1
**Empirical Novelty And Significance:** 1
**Recommendation:** 3

**Strength And Weaknesses:**

Strengths:

+ Apply frequency decomposition for the holistic statistics-based style transfer method.
+ Building the relationship between content loss (&& Gram matrix) and amplitude (&& phase) of Fourier transform.


Weakness:
+ The designed method in this paper is completely based on the holistic style statistics hypothesis, which limits its performance. The authors propose two operations to enhance the content structure of stylized results and interpolate between $F_{cs}$ and $F_c$ respectively. However, it is still at the cost of balancing the style information and content structures. See point 2 for details. So I would say this contribution to the style transfer task seems to be limited.
+ From Fig. 1, the results of this method are weaker than those of WCT regarding the style textures; for example, the color of the content image (e.g. green regions) are preserved on the $3^{rd}$ and $5^{th}$ columns. Therefore, does it mean that the phase replacement operation plays a negative effect on the preservation of style features? This can also be reflected in Fig. 2. I assume that this is connected to the fact that Eq. (8) contains a cosine term with different channel features, and the phase replacement operation will weaken this relationship, thereby damaging the style characteristics in the results.
+ Patch based style transfer methods (e.g. SANet) and methods combining patch alignment and holistic statistics matching (e.g. AvatarNet), also aim for the issues of spatial structure distortion, and thus can maintain a good content structure. The authors should make comparisons with such methods.
+ This submission mainly depends on the qualitative comparisons, which might be very subjective. I suggest to use some widely used quantitative metrics, like perceptual loss, to evaluate the capability in capturing style patterns. At least, the authors should perform some user studies over randomly generated results.
+ As far as I know, the running speed of linear WCT is comparable to that of AdaIN. However, in Tab. 1, the running speed of linear WCT is four times more than that of WCT, and even fourteen times more than that of AdaIN, which is inconsistent with other papers.
+ How to enhance the stylistic features in synthesized results is the core issue of style transfer. It would be better, if the authors can improve it from the perspective of Fourier transform. For instance, the authors can weigh the $F_{cs}$ and $F_c$ with different frequency characteristics for trade-off. I look forward to more interesting frequency-based manipulation strategy for stylized images.
+ There is still no user study result.
+ How to do the quantitative experiments is also unclear. For example, how to select the input content && style images? How to select the synthesized images to compute metrics? Can the authors release their test data for later reproduction? What are the resolutions of test images?
+ The selected representative images look too similar to LinearWCT. And its running speed is also unconvincing. I ran many experiments with this method before and it is even faster than AdaIN. Considering the limited visual performance, I would say the empirical improvement is very trivial.
+ Some other experiment problems: e.g. the compared methods are a bit out-of-date; More state-of-the-art methods should be compared with, like MCCNet [1] and StyTr2 [2].

[1] Arbitrary Video Style Transfer via Multi-Channel Correlation, AAAI 2021.

[2] StyTr2:Unbiased Image Style Transfer with Transformers, CVPR 2022.

====== My last NeurIPS comments ======

Thank the authors for their further feedback with details! After reading the response, however, I would like to preserve my opinions.

With the updated running time, it can be seen that LinearWCT is ~25 faster than the proposed one and having comparable (even better) results. It is very hard to convince me to give a positive rate.

Thus, I encourage the authors to improve their paper by incorporating the suggestions from all the reviewers into their revisions.


====== Other comments ======

Here I do not plan to append more comments from other NeurIPS reviewers now, but I do agree with some of their feedbacks including concerns about the technical novelty, limited experimental improvements and lack key evaluation results. Especially for the technical novelty and limited experimental improvements, all the NeurIPS reviewers have serious concerns. Thus I think these issues should be addressed before acceptance.

**Summary Of The Paper:**

This paper analyzes the global style transfer method from the perspective of Fourier analysis. Specifically, the formula of style transfer is converted into Fourier transform under the umbrella of global statistics. Thus the relationships between Fourier phase (Fourier amplitude) and content loss (Gram matrix) are achieved. In this way, the authors design the phase replacement operation to improve the preservation of content structure and the frequency combination operation for content & style feature interpolations with different frequency components.

**Summary Of The Review:**

The reviewer has reviewed this paper in NeurIPS 2022, and it was rejected by all reviewers. It is surprising that the authors did not address any of my main comments (e.g. the technical novelty issue, trivial experimental improvement and lack key evaluation results), but directly recycle it to ICLR 2023 by just doing some minor revisions. But as I mentioned during NeurIPS review, some main concerns need to be addressed before acceptance. And I think I can do nothing if the authors decide to ignore my (and other reviewers') suggestions. Thus I will also copy my previous review from NeurIPS 2022 console and paste it here. I strongly encourage the authors to incorporate the suggestions from all the NeurIPS reviewers into their revisions.

---

> ### Author Response · Authors · 2022-12-05
> **Thanks for your constructive feedback!**
>
> We thank the reviewer for your time and thoughtful comments. We will take the suggestions and improve our experiments. We would like to discuss about your concerns especially in the technical novelty of our paper.
>
> - In the comments, the reviewer mentions a concept called `holistic statistics-based style transfer` and claims corresponding methods have limited performance. Could the reviewer please provide more information and papers on its definition and on its inferiority?
> - Since the optimal point sets of content loss and those of style loss are non-intersect, **all** the style transfer methods are actually balancing the style information and content structure. We believe it does contribute to the community if we could find out how to balance them appropriately. Additionally, the balancing operations we propose could be applied to multiple methods in a plug-and-play manner.
> - Based on the following theoretical analysis, we consider the influence of the cosine term as **limited**. Please note that the $c^{th}$ element on the diagonal of Gram matrix $diag\(FF^{\top}\)$ equals $ \sum_{k}F_{c, k}^{2} = \mu_{c}^{2} + \sigma_{c}^{2}$, where $\mu$ and $\sigma$ are the channel-wise mean and the standard variance of feature maps $F$. As Phase Replacement (PR) keeps the amplitude of stylized feature maps $F_{cs}$, elements on the diagonal are unchanged, which means $(\mu^{cs})^{2} + (\sigma^{cs})^2 = (\mu^{s})^{2} + (\sigma^{s})^{2}$ still holds. Since $\mu^{cs} = \mu^{s}$ , the non-negative channel-wise standard variances are theoretically the same $\sigma^{cs} = \sigma^{s}$. As a result, there are no theoretical changes caused by PR in the scope of the style loss $\mathcal{L}_{style} = \|\|\mu^{cs} - \mu^{s}\|\|_2 + \|\|\sigma^{cs} - \sigma^{s}\|\|_2$, which is widely used in SANet and AdaIN. In this way, the on-diagonal elements are quite **sufficient** to represent the main style characteristics of output results.

---

### Official Review · Reviewer_vcHP · 2022-10-24

**Confidence:** 5
**Correctness:** 3
**Technical Novelty And Significance:** 2
**Empirical Novelty And Significance:** 2
**Recommendation:** 3

**Clarity, Quality, Novelty And Reproducibility:**

The paper is okay and has every component to reproduce the work. However, the main argument needs to be supported with more experiments in various perspectives in a more aggressive way.

**Strength And Weaknesses:**

Pros
* This paper provides a new interpretation based on the frequency analysis.
* The paper is written clearly and idea is reasonable.

Cons
* The frequency combination (FC) does not seem to give good stylization effect. It is well-known that the phase information contains the structural information of an image. In style transfer, one may want to have strong strokes or textures. However, since the proposed method replace the phase information directly, this is hard to be done by design. To me, this method seems to work better for photorealistic stylization, not artistic.

* Even the metrics shown in Table 1 is using SSIM, which is the metric for checking structural preservation, not perceptual quality. If this method is meant for artistic style transfer, please measure some perceptual metrics such as GRAM difference to the results of the original optimization method and user study. That said, please perform user studies asking various aesthetic aspects using randomly generated results.
* Missing comparison to relevant work. The problems of spatial structure distortion are also addressed by SANet and AvatarNet, which are reported to preserve content structure pretty well with nice stylization effect. The writers ought to contrast their approaches with similar ones.
* Overall, the results seem to be very weak. For example, the authors say that the FC gives better controllability and stylization by manipulating them in frequency domain (eq. 12). However, in my opinion, this gives more unnatural effect at the edges (e.g., boundaries of buildings, Figure 10) compared to the linear combination. Where to apply styles in terms of frequency difference seems not to be a good choice since content of an image is a more entangled entity that cannot be detached merely by frequencies.

Minor
Line 58, Next, We -> Next, we



**Summary Of The Paper:**

Based on Fourier analysis for existing universal style transfer (UST) approaches, this paper demonstrates that existing methods handle all frequency components identically, with the exception of the zero-frequency component. They also show that the whitening and coloring transform (WCT) may affect phase information whereas AdaIN does not. This backs up the empirical finding that AdaIN retains its structural integrity better than WCT. They suggest replacing the phase information for better structure preservation based on this research.

**Summary Of The Review:**

Overall, the idea of the paper is interesting, but further research is needed.

---

> ### Author Response · Authors · 2022-12-05
> **Thanks for your constructive feedback!**
>
> We thank the reviewer for your time and thoughtful comments. We would like to respond your concerns point by point.
>
> **Q1:**  *The frequency combination (FC) does not seem to give good stylization effect... In style transfer, one may want to have strong strokes or textures. However, since the proposed method replace the phase information directly, this is hard to be done by design. To me, this method seems to work better for photorealistic stylization, not artistic.*
>
> **A1:** Thanks for the helpful suggestions and we will take them to improve the paper. While in terms of the limitations of phase replacement (which we call PR instead of FC as you mentioned), one of our core ideas is to cooperate it with various style transform algorithms in a plug-and-play manner. In this way, users could choose not to replace the phase when they want strong strokes or textures and handle it all over to the specific style transform algorithm. The proposed PR works as a universal operation instead of a specific method. Thus we believe the weakness mentioned above should be limited because we offer a universal option for different methods instead of a specific method unable to handle strong strokes.
>
> **Q2:** *Even the metrics shown in Table 1 is using SSIM, which is the metric for checking structural preservation, not perceptual quality. If this method is meant for artistic style transfer, please measure some perceptual metrics such as GRAM difference to the results of the original optimization method and user study. That said, please perform user studies asking various aesthetic aspects using randomly generated results.*
>
> *Missing comparison to relevant work. The problems of spatial structure distortion are also addressed by SANet and AvatarNet, which are reported to preserve content structure pretty well with nice stylization effect. The writers ought to contrast their approaches with similar ones.*
>
> **A2:** Thanks for the helpful suggestions. We acknowledge that we need to provide multiple metrics and comparisons to relevant work. We will work on it to improve the paper.
>
> **Q3:** *Overall, the results seem to be very weak. For example, the authors say that the FC gives better controllability and stylization by manipulating them in frequency domain (eq. 12). However, in my opinion, this gives more unnatural effect at the edges (e.g., boundaries of buildings, Figure 10) compared to the linear combination. Where to apply styles in terms of frequency difference seems not to be a good choice since content of an image is a more entangled entity that cannot be detached merely by frequencies.*
>
> **A3:** Thanks for the feedback. For the "unnatural" issue, since we are in the context of artistic style transfer, we believe the problem might be limited. We agree that content cannot be detached merely by frequencies, while our main idea is to stylize the details or the general arrangement of images instead of the content. Additionally, we would like to state that frequency combination could simply realize linear combination by setting $\alpha(u, v) = a$ , where $a$ is the factor of linear combination.

---

### Author Response · Authors · 2022-12-02
**Thanks for the thoughtful and helpful feedback of all reviewers!**

We thank all of the reviewers of their thoughtful and helpful feedback. Because of personal affairs, we are running behind the schedule and we apologize for late response. We will keep track of feedback to the paper and we are open to any suggestions or challenges for the improvement of the paper.

---

### Decision · Program_Chairs · 2023-01-20

**Decision:**

Reject

**Justification For Why Not Higher Score:**

Major shortcomings in evaluation, not to mention concerns with fundamental properties of the method (impossible to assess without improved evaluation)

**Justification For Why Not Lower Score:**

N/A

**Metareview: Summary, Strengths And Weaknesses:**

It seems clear that the paper needs a finish revision, including in particular the need to address numerous shortcomings in evaluation. The authors acknowledge this in the discussion. I agree with the majority of reviewers that this submission is not ready at the moment for acceptance.